# Recent shifts in the genomic ancestry of Mexican Americans may alter the genetic architecture of biomedical traits

Melissa L Spear[1,2,3,4]*, Alex Diaz-Papkovich[3,5], Elad Ziv[6,7,8,9], Joseph M Yracheta[10,11], Simon Gravel[3,4], Dara G Torgerson[3,4,12], Ryan D Hernandez[2,3,4,8,13,14]*

[1]Biomedical Sciences Graduate Program, University of California, San Francisco, San Francisco, United States; [2]Department of Bioengineering and Therapeutic Sciences, University of California, San Francisco, San Francisco, United States; [3]McGill Genome Centre, McGill University, Montreal, Canada; [4]Department of Human Genetics, McGill University, Montreal, Canada; [5]Quantitative Life Sciences Program, McGill University, Montreal, Canada; [6]Division of General Internal Medicine, University of California, San Francisco, San Francisco, United States; [7]Department of Medicine, University of California, San Francisco, San Francisco, United States; [8]Institute of Human Genetics, University of California, San Francisco, San Francisco, United States; [9]Helen Diller Family Comprehensive Cancer Center, University of California, San Francisco, San Francisco, United States; [10]Native BioData Consortium, Eagle Butte, United States; [11]Bloomberg School of Public Health, Johns Hopkins University, Baltimore, United States; [12]Department of Epidemiology and Biostatistics University of California, San Francisco, San Francisco, United States; [13]Bakar Computational Health Sciences Institute, University of California, San Francisco, San Francisco, United States; [14]Quantitative Biosciences Institute, University of California, San Francisco, San Francisco, United States

*For correspondence:
mlspear09@gmail.com (MLS);
ryan.hernandez@me.com (RDH)

**Competing interests:** The authors declare that no competing interests exist.

**Abstract** People in the Americas represent a diverse continuum of populations with varying degrees of admixture among African, European, and Amerindigenous ancestries. In the United States, populations with non-European ancestry remain understudied, and thus little is known about the genetic architecture of phenotypic variation in these populations. Using genotype data from the Hispanic Community Health Study/Study of Latinos, we find that Amerindigenous ancestry increased by an average of ~20% spanning 1940s-1990s in Mexican Americans. These patterns result from complex interactions between several population and cultural factors which shaped patterns of genetic variation and influenced the genetic architecture of complex traits in Mexican Americans. We show for height how polygenic risk scores based on summary statistics from a European-based genome-wide association study perform poorly in Mexican Americans. Our findings reveal temporal changes in population structure within Hispanics/Latinos that may influence biomedical traits, demonstrating a need to improve our understanding of admixed populations.

## Introduction

The United States Census Bureau refers to the Hispanic/Latino ethnicity as a category for individuals who self-identify as 'a person of Cuban, Mexican, Puerto Rican, South or Central American, or other Spanish culture or origin regardless of race (*United States Government, Executive Office of the*

*President, Office of Management and Budget, Office of Information and Regulatory Affairs, 1997*). As such, this broad ethnic group living in the United States is a culturally, phenotypically, and genetically diverse continuum of populations. Individuals who identify as Hispanic/Latino have varying proportions of Amerindigenous, African, and European genetic ancestries, each with their own unique continental demographic history. Demographic forces such as population bottlenecks, expansions, and migration as well as adaptation to novel environments resulted in observable differences in continental patterns of genetic variation (*Nelson et al., 2008*; *Abecasis et al., 2012*; *Auton et al., 2015*). These differing patterns were shaped by many historical events of migration which included the founding of the Americas by Amerindigenous populations, the colonization by Europeans, and the African slave trade (*Gravel et al., 2013*; *Homburger et al., 2015*; *Moreno-Estrada et al., 2014*; *Moreno-Estrada et al., 2013*; *Reich et al., 2012*; *Bryc et al., 2015*; *Conomos et al., 2016*; *Han et al., 2017*; *Baharian et al., 2016*; *Jordan et al., 2019*; *Micheletti et al., 2020*). However additional complexities surrounding these events remain highly understudied.

Demographic history has shaped the genetic architecture of modern human phenotypic variation (*Agarwala et al., 2013*; *Eyre-Walker, 2010*; *Maher et al., 2013*; *Simons et al., 2014*; *Uricchio et al., 2016*; *Yang et al., 2015*), and is critical to consider in the search for the genetic basis of complex diseases. The demography of the United States has changed drastically over the 20th century, and by 2044 is predicted to become a 'minority-majority' country whereby no one racial/ethnic group comprises more than 50% of the population. By 2060 Hispanics/Latinos are projected to make up 29% of the US population or 119 million individuals (*Colby and Ortman, 2015*). However, to date, population-based medical genomics research [and its subsequent benefits, including polygenic risk score (PRS) profiling] have been disproportionately focused on individuals of European descent, with the findings primarily benefiting European populations (*Bustamante et al., 2011*; *Martin et al., 2019*). Despite the increases in sample sizes, rates of discovery, and traits studied, Hispanic or Latin American participation in genome-wide association studies (GWAS) has continued to hover around 1% (*Popejoy and Fullerton, 2016*; *Mills and Rahal, 2019*). This trend, along with factors ranging from research abuse and community mistrust to community superstition and apathy have led to a situation where these populations (and other non-European populations) are particularly vulnerable to falling behind in receiving the benefits of the precision medicine revolution (*Martin et al., 2019*; *Popejoy and Fullerton, 2016*).

In this study, we utilize the largest genetic study of Hispanics/Latinos in the U.S. to date – the Hispanic Community Health Study/Study of Latinos (HCHS/SOL) (*Conomos et al., 2016*) – to understand how patterns of genetic variation in Hispanic/Latino populations in the United States have changed over the last century, and evaluate the impact such changes may be having on complex traits.

## Results

### Global ancestry proportions among HCHS/SOL Hispanic/Latino Populations

Using the subset of sites that overlapped with our African, European, and Amerindigenous reference panels, we called 3-way global ancestry estimates for 10,268 unrelated HCHS/SOL individuals (see Materials and methods). *Figure 1A* summarizes the global ancestry proportions shaded by admixture estimates in a ternary plot, recapitulating the original HCHS/SOL analysis of continental ancestry (*Conomos et al., 2016*). However, while several population groups appear to have overlapping ancestry proportions (*Figure 1B*), this analysis masks more subtle structure in subcontinental ancestry. To investigate subtle population structure across these self-identified population groups, we performed UMAP on the top three principal components (see Materials and methods and *Figure 1—figure supplement 1C*), and find substantial structure across self-identified groups (*Figure 1C–D*). We find that Dominicans, who have the highest average proportions of African ancestry, are in the middle, with Puerto Ricans and Cubans, diverging in opposite directions (*Figure 1D*) with clines of increasing European ancestry proportions (*Figure 1C*). Further, while self-identified Mexican, Central, and South American groups appear to have overlapping ancestry proportions in *Figure 1A–B*, UMAP represents the Mexican Americans and Central/South American groups as large, separate

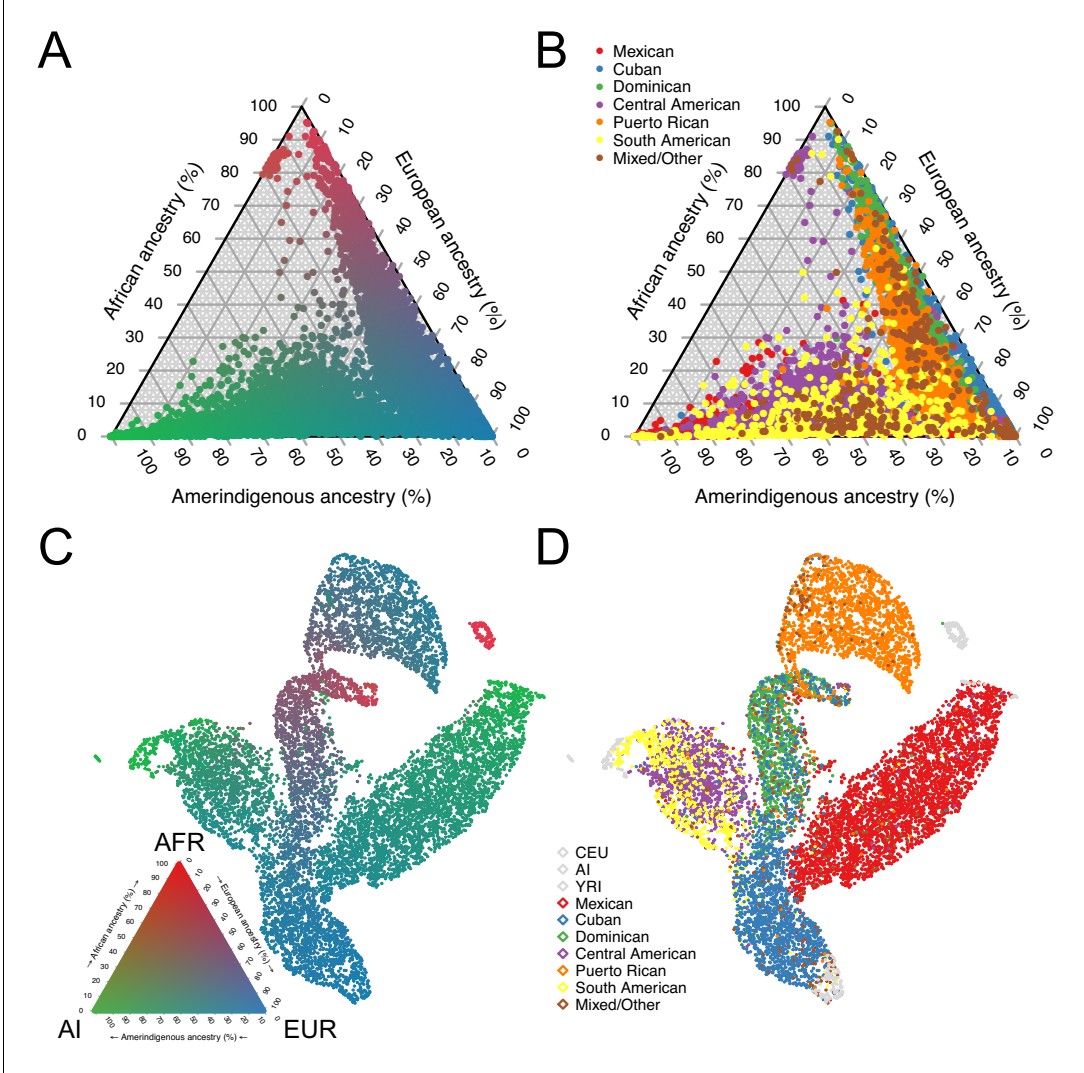

**Figure 1.** Genomic ancestry and population structure in HCHS/SOL. (**A**) Ternary plot of HCHS/SOL (n = 10,268) colored by admixture proportions. (**B**) Ternary plot of global ancestry proportions colored by population for 10,268 HCHS/SOL individuals (**C**) Uniform Manifold Approximation and Projection (UMAP) plot depicting the genetic diversity of HCHS/SOL and the reference panel (n = 10,591) using three principal components, colored by admixture proportions Within the legend, AFR, EUR, and AI refer to African, European, and Amerindigenous global ancestries, respectively. (**D**) UMAP plot of HCHS/SOL and the reference panel (n = 10,591) using three principal components, colored by HCHS/SOL population.

The online version of this article includes the following figure supplement(s) for figure 1:

**Figure supplement 1.** Ancestral diversity of HCHS/SOL populations.

wings that diverge from self-identified Cubans and Dominicans, with both clusters diverging with clines of increasing ancestry toward different Amerindigenous (AI) populations (*Figure 1C–D* and *Figure 1—figure supplement 1B*). When we included multiple European and African reference populations in our analyses as well as without reference populations, UMAP maintained the representation of separate clusters for each of the HCHS/SOL populations (*Figure 1—figure supplement 1C–G*). These clusters with varying AI ancestries are consistent with *Conomos et al., 2016*, however the UMAP embedding consolidates the signal present in the top three PCs into a succinct two-dimensional visualization.

## Dynamic global ancestry proportions in Mexican Americans

For each of the HCHS/SOL populations, we evaluated differences in global ancestry estimates over time while accounting for the sampling method (referred to as 'sampling weight', see Materials and methods) used for the design of the HCHS/SOL study (*Sorlie et al., 2010*). We found that in all populations, the effect size for AI ancestry on birth year is positive, though only statistically significant after multiple testing in the Mexican American ($\beta$=0.0023; 95% CI:0.0021–0.0025, p=3.58E-22; *Figure 2A–B*) and Central American ($\beta$=0.0013; 95% CI:0.0009–0.0017, p=0.0013) cohorts (*Supplementary file 1*). Due to the larger sample size, magnitude of the effect, and statistical significance, we shift our focus to Mexican Americans. In Mexican Americans, the increase in AI global ancestry over time was consistent across multiple data stratifications including recruitment region, US-born or not US-born, educational attainment, and gender (*Table 1* and *Supplementary file 2*), and was robust to alternative methods for estimating global ancestry proportions (e.g. based on the summation of RFMix local ancestry estimates; *Figure 2—figure supplements 1* and *2*). We identified significant differences in AI ancestry between recruitment region (t-test, 95% CI:0.12–0.15, p<2.2E-16), US-born or not US-born individuals (t-test, 95% CI:0.06–0.09, p<2.2E-16), and educational attainment, which can be considered a proxy for socioeconomic status (one-way ANOVA, p<2E-16). In order to further assess changes in global ancestry distributions over time, we performed bootstrap resampling over individuals (n = 1000) of global AI ancestry for the Mexican Americans. We observed a consistent increase in AI ancestry with fitted locally estimated scatterplot smoothing (LOESS; *Figure 2B*) when individuals were binned by birth year decades (*Figure 2—figure supplement 3*). On average, global AI ancestry has increased ~20% over the 50 year period for Mexican Americans born from 1940 to 1990.

We replicated the increase in global AI ancestry over time in a smaller, independent cohort of self-identified Mexican Americans (n = 705) from the Health and Retirement Study (HRS) (*Fisher and*

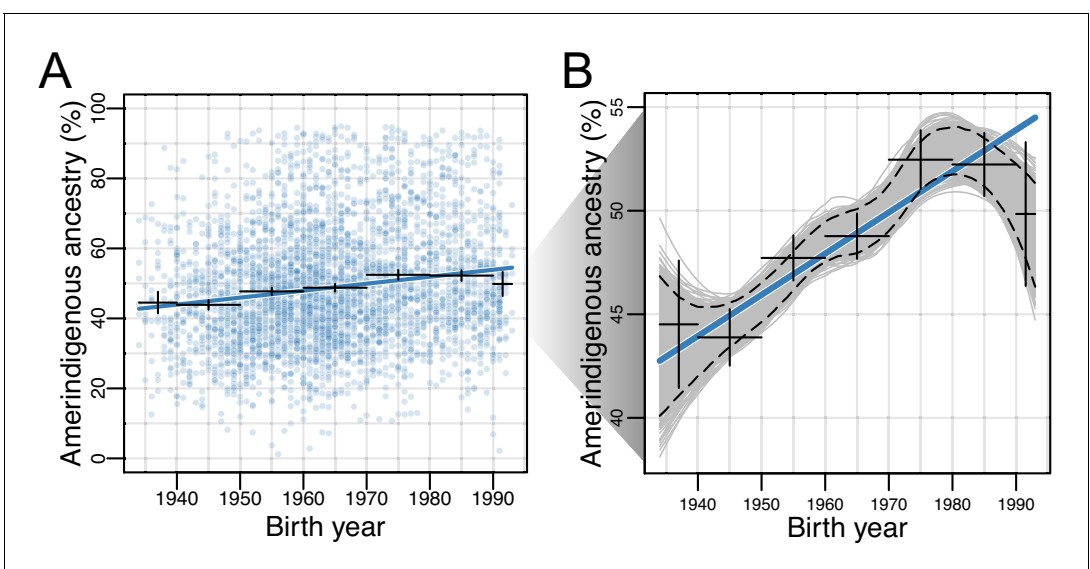

**Figure 2.** Amerindigenous ancestry has increased over time in Mexican Americans. (**A**) Global Amerindigenous ancestry proportions plotted by birth year for Mexican Americans (n = 3,622). Fitted line is multiple regression of Amerindigenous ~ birth year + sampling weight. Bars represent 95% confidence intervals for individuals grouped by decade. (**B**) Bootstrap resampling (n = 1000 iterations) of Amerindigenous global ancestry for the Mexican American individuals with a fitted LOESS curve for each iteration. Dashed lines represent the 95% quantile range of LOESS curves and the blue line represents the fitted regression line from A.

The online version of this article includes the following figure supplement(s) for figure 2:

**Figure supplement 1.** Concordance of ADMIXTURE and RFMix global ancestry estimates.

**Figure supplement 2.** Amerindigenous ancestry has increased over time in Mexican Americans.

**Figure supplement 3.** Distributions of Amerindigenous global ancestry means for HCHS/SOL Mexican Americans (n = 3622) generated by 1000 bootstrap resampling iterations within each decade of binned birth years.

**Figure supplement 4.** Replication in the Health and Retirement Study for 705 self-identified Mexican Americans.

**Figure supplement 5.** The increase in estimated AI ancestry over time is conditional on the number of US-born parents.

**Table 1.** Relationship of Amerindigenous global ancestry and birth year for Mexican Americans stratified by recruitment region, US-born vs non-US-born status, gender and educational attainment.

For recruitment region, data stratification was limited to Chicago and San Diego as sample size for the Bronx and Miami was limited: 124 and 25 individuals, respectively. Education attainment was categorized as either less than a high school diploma or equivalent degree (<HS), equal to a high school diploma or equivalent degree (=HS), or post-secondary education (>HS). The significance threshold was set at 0.006 using Bonferroni correction for multiple testing (0.05/9).

| Category | N | Mean | Median | R2 | Effect | Std.err | p |
|---|---|---|---|---|---|---|---|
| All | 3622 | 0.489 | 0.468 | 0.027 | 0.0023 | 0.0002 | 3.58E-22 |
| Chicago | 1310 | 0.562 | 0.550 | 0.017 | 0.0016 | 0.0005 | 0.0006 |
| San Diego | 2163 | 0.428 | 0.422 | 0.012 | 0.0012 | 0.0002 | 4.29E-07 |
| US-born | 634 | 0.427 | 0.418 | 0.063 | 0.0027 | 0.0004 | 1.77E-10 |
| Non US-born | 2987 | 0.502 | 0.481 | 0.050 | 0.0032 | 0.0003 | 1.38E-30 |
| Male | 1500 | 0.494 | 0.475 | 0.038 | 0.0028 | 0.0004 | 3.83E-14 |
| Female | 2122 | 0.485 | 0.462 | 0.022 | 0.0019 | 0.0003 | 3.07E-10 |
| <HS | 1518 | 0.520 | 0.500 | 0.045 | 0.0026 | 0.0004 | 1.39E-12 |
| = HS | 960 | 0.501 | 0.479 | 0.022 | 0.0018 | 0.0005 | 0.0003 |
| >HS | 1140 | 0.436 | 0.422 | 0.045 | 0.0027 | 0.0004 | 6.53E-13 |

*Ryan, 2018*). The HRS Mexican Americans are older compared to the HCHS/SOL Mexican Americans (birth year distribution: 1915–1981; mean = 1943, median:1942) and have lower levels of global AI ancestry on average (mean = 0.29), but we still observed an increase in global AI ancestry over time ($\beta$=0.00082; 95% CI: 0.0005–0.0012; p=0.02; *Figure 2—figure supplement 4A*). We performed 1000 bootstrap resampling iterations of the linear regression model (global AI ancestry ~birth year) fitted to the data. From these resampling iterations, 98.2% of the tests had a slope >0% and 61.5% of the regression p-values were less than 0.05 (*Figure 2—figure supplement 4B–4D*).

A previous study (*Baharian et al., 2016*) identified ancestry biased migration in African Americans where individuals with higher proportions of European ancestry migrated first out of the South during the Great Migration followed by individuals with higher proportions of African ancestry. We hypothesized that a similar process occurred in US Hispanic/Latino populations, whereby earlier immigrants to the US had higher proportions of European ancestry followed by recent immigrants having higher proportions of global AI ancestry. In our non-US-born individuals (N = 2987), we evaluated differences in ancestry estimates over time while accounting for years in the US and sampling weight and identified a significant effect of years in the US ($\beta$=−0.0009; 95% CI: −0.0012, −0.0006; p=0.0006) suggesting that individuals who arrived earlier to the US had less AI ancestry. However, accounting for this did not change the effect of birth year on the proportion of global AI ancestry ($\beta$=0.0028; 95% CI: 0.0025–0.0031; p<2E-16) suggesting that ancestry biased migration does not fully explain the dynamic AI ancestry patterns we have inferred.

For US-born individuals we assessed whether parental birthplace could explain the increases in global AI ancestry. Of the 634 US-born individuals, 385 had parents both born outside of the US, 149 had one parent born outside of the US, and 97 had both parents born within the US. We tested a model with an interaction between estimated birth year and the number of parents born in the United States. We found a strong positive relationship between estimated birth year and increase in AI ancestry for those with both parents born outside the US, who formed the baseline group in this model ($\beta$=0.004; 95% CI: 0.0034-0.0046; p=4.85e-12) (*Figure 2—figure supplement 5*). The relationship between estimated birth year and AI ancestry for those with one parent born in the US was still positive but smaller when the effect size was added to the baseline mean ($\beta$=-0.0034; 95% CI: -0.0043, -0.0025; p=0.000123) and for those with both parents born in the US the relationship was overall negative ($\beta$=-0.0049; 95% CI: -0.006, -0.0038; p=1.04e-5).

## Little evidence for subcontinental population structure

We explored whether the increase in global AI ancestry over time could occur in tandem with local changes in the specific subcontinental AI ancestries over time. If it were the case, then we would

expect subtle signals of genetic divergence in AI ancestry tracts over time. To investigate this, we calculated $F_{ST}$ within AI ancestry tracts between all pairs of birth-decades (see Materials and methods). *Figure 3—figure supplement 1* shows all pairwise comparisons among birth-decades, and demonstrates that while the estimates of $F_{ST}$ are negligible (with many estimates below 0), there is a subtle trend of increasing $F_{ST}$ as birth-decade differences increase (though individuals born in the 80 s and 90 s show a conflicting pattern).

We further investigated patterns of subcontinental population structure using genetic diversity, $\pi$, in AI ancestry tracts for each birth-decade (see Materials and methods). We hypothesized that if there were increased migration from multiple AI source populations (coupled with rapid population growth in Mexican American communities), then genetic diversity should be increasing over time. We found the opposite: *Figure 3A* shows a subtle decrease in genetic diversity ($\pi$) over time from the 1930s to the 1980s in non-US-born Mexican Americans, and a subtle decrease in US-born Mexican Americans from the 70 s to the 90 s (while remaining roughly constant from the 30 s to the 70 s).

## AI ancestry tract lengths have not changed, but runs of homozygosity (ROH) have increased

If there was a rapid increase in the migration of individuals with high AI ancestry, we would expect to see an increase in long AI tracts over time. To test this, we calculated the length of each RFMix inferred local ancestry tract in each Mexican American individual and tested for differences in the distribution of tract lengths across birth-decades using a multiple linear regression model (see Materials and methods). We found no significant associations between the decade bin and the proportion of AI ancestral tracts at various lengths (*Figure 3B*; $\beta = 0.04$, CI = $(-0.019–0.099)$; p=0.19), even when testing for violations of model assumptions (e.g. normalizing the tracts per bin by the number of individuals, or excluding the 1930s and/or 1990s individuals due to the small sample size in each bin).

While there are no statistical differences in the length of admixture tracts, it is possible that local ancestry tracts have accumulated in specific regions of the genome to drive the increased global ancestry proportions over time. We used local ancestry estimates generated across the genome to perform admixture mapping in HCHS/SOL Mexican Americans to determine if younger individuals harbored excess AI ancestry in certain regions of the genome. Although we tried two different models (see Materials and methods), we did not find any loci to be significantly associated with birth year across the genome (*Figure 3—figure supplement 2*).

We find that there are no changes in AI ancestry tract lengths over time nor any regions of the genome that seem to be accumulating AI ancestry at disproportionate rates, yet genetic diversity has decreased over time in the AI ancestry tracts of Mexican Americans despite rapid growth of the census population size. We therefore investigated whether this population has experienced increased haplotype homozygosity over time. We investigated this possibility by exploring runs of homozygosity (ROH) across the genomes of each of the 3622 Mexican Americans. We classified ROH into three categories: short, medium, and long, based on the length distribution in the population. Generally, short ROH are tens of kilobases in length and likely reflect the homozygosity of old haplotypes; medium ROH are hundreds of kilobases in length and likely reflect background relatedness in the population; and long ROH are hundreds of kilobases to several megabases in length and are likely the result of recent parental relatedness. Overall, we find a significant positive correlation between birth year and the total ROH (summed across size classes; $\tau = 0.0449$, p=6.12e-5, Kendall's rank correlation), but this signal becomes stronger when we restrict our analysis to ROH calls that overlap AI ancestry tracts ($\tau=0.065$, p=7.39e-9). *Figure 3C* shows a fitted LOESS curve to the proportion of the genome with AI (or European) ancestry covered by ROH across the genomes of Mexican Americans as a function of their birth year, broken down by ROH size class (see *Figure 3—figure supplement 3* for the distribution of ROH by length classes and ancestry). When stratified by size class and normalized by AI global ancestry, the associations (all Kendall's rank correlation) in AI ROH were primarily driven by the short ($\tau=0.097$, p<2.2E-16), and medium ($\tau=0.084$, p=1.27E-13) size classes (while long ROH was insignificant after multiple testing due to the small number of long ROH across individuals; $\tau = -0.032$, p=0.004). We observed the opposite pattern when ROH were restricted to European ancestry segments of the genome: there is a significant negative correlation between birth year and the total ROH that overlap European ancestry tracts ($\tau=-0.089$, p=1.82E-14).

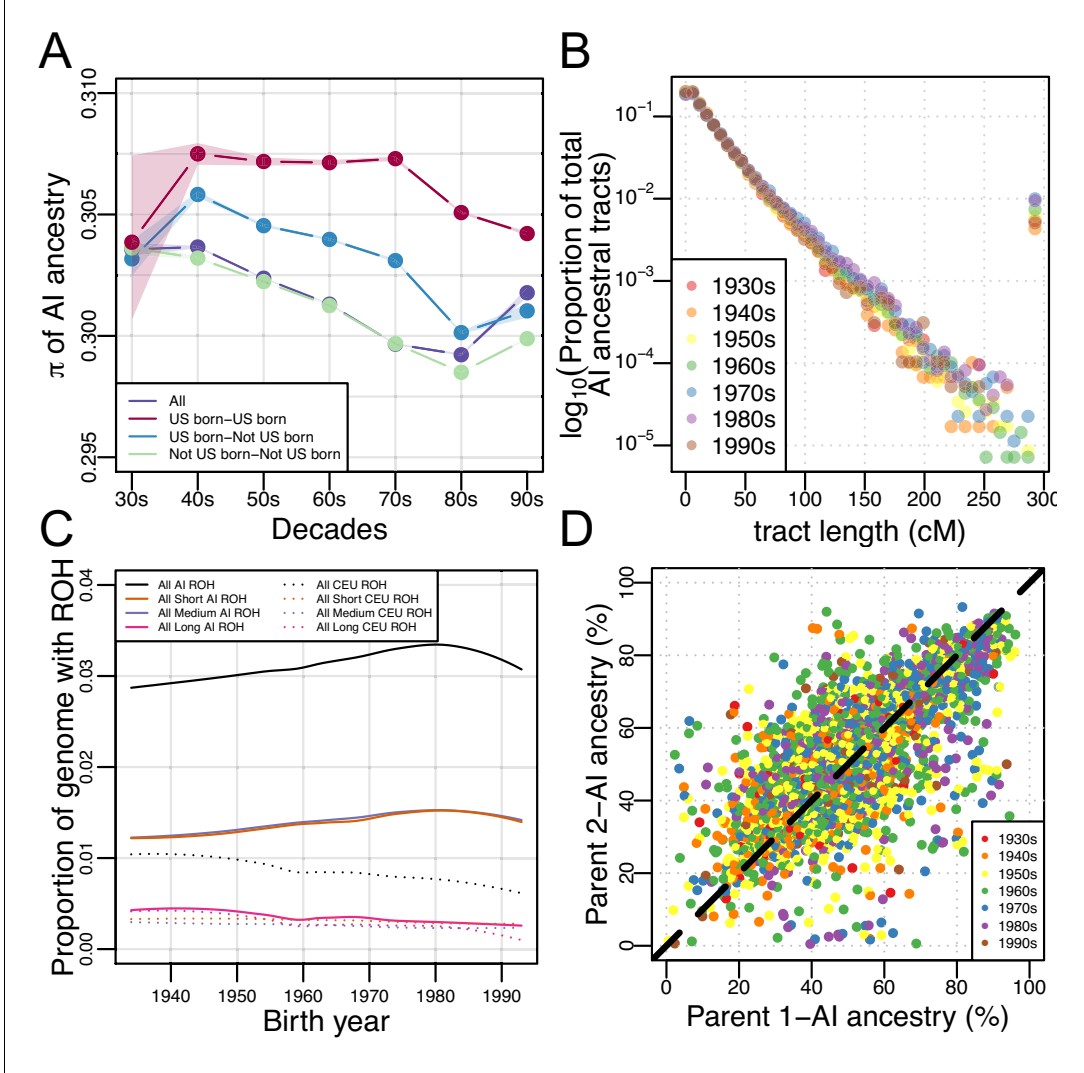

**Figure 3.** Architecture of genetic diversity in Mexican American Genomes. (**A**) Genetic diversity (π) in Amerindigenous ancestry tracts stratified by US-born/not US-born status, and calculated between pairs of individuals born within each decade (with shaded envelopes showing 95% confidence intervals for each group). (**B**) Proportion of total Amerindigenous (AI) ancestral tracts in the HCHS/SOL Mexican American population by decade. (**C**) Variation in ROH by birth year. Solid lines show LOESS of the proportion of the genome with AI ancestry that overlap ROH of different lengths, while dotted lines show LOESS of the proportion of the genome with European ancestry that overlap ROH of different lengths. (**D**) Scatter plot of parents' inferred global Amerindigenous (AI) ancestries using ANCESTOR.

The online version of this article includes the following figure supplement(s) for figure 3:

**Figure supplement 1.** $F_{ST}$ within Amerindigenous ancestral tracts.

**Figure supplement 2.** Admixture mapping in HCHS/SOL Mexicans (n = 3622) for Amerindigenous ancestry and (**A**) birth year and (**B**) generation.

**Figure supplement 3.** Runs of homozygosity (ROH) in HCHS/SOL Mexican Americans.

**Figure supplement 4.** Ancestry-related assortative mating in HCHS/SOL Mexican Americans.

**Figure supplement 5.** Standard neutral model simulations result in no change in ancestry proportions over time.

**Figure supplement 6.** Population growth does not affect the mean ancestry proportions in a population.

**Figure supplement 7.** Ancestry-based assortative mating does not change mean ancestry proportions, though variance in ancestry proportions can increase.

**Figure supplement 8.** Ancestry-based fecundity differences can induce systematic changes in ancestry proportions in a population.

**Figure supplement 9.** The ancestry proportions in the migrant population are modeled as a Beta distribution, with mean given by a weighted average between the domestic population at one with weight *mAI*.

**Figure supplement 10.** Simulating the effects of migration on changing ancestry proportions.

**Figure supplement 11.** Similar to *Figure 3—figure supplement 10*, but adding assortative mating (*AM* = 0.75, consistent with our data) and ancestry-based fecundity differences (*FAI* = 0.1, see *Figure 3—figure supplement 8A*).

*Figure 3 continued on next page*

*Figure 3 continued*

**Figure supplement 12.** Similar to *Figure 3—figure supplement 10*, but adding assortative mating ($AM = 0.75$, consistent with our data) and ancestry-based fecundity differences ($FAI = 0.2$, see *Figure 3—figure supplement 8A*).

**Figure supplement 13.** Similar to *Figure 3—figure supplement 10*, but adding assortative mating ($AM = 0.75$, consistent with our data) and ancestry-based fecundity differences ($FAI = 0.4$, see *Figure 3—figure supplement 8A*).

**Figure supplement 14.** Similar to *Figure 3—figure supplement 10*, but adding assortative mating ($AM = 0.75$, consistent with our data) and ancestry-based fecundity differences ($FAI = 0.8$, see *Figure 3—figure supplement 8A*).

## Strong ancestry-related assortative mating in HCHS/SOL Mexicans

Given that short and medium length ROH have increased over time, it appears that background relatedness within AI ancestry in Mexican Americans has increased over time (but not an increase in recent parental relatedness). One way for this to occur is if individuals with similar ancestry patterns tend to mate with one another more often than expected under a model of random mating (i.e. assortative mating). To measure assortative mating, we estimated the ancestral proportions of the biological parents of each HCHS/SOL Mexican American (see Materials and methods). With individuals from all decades pooled together, we found the inferred biological parental AI ancestries to be significantly correlated (*Figure 3D*, r = 0.708, 95% CI:0.69–0.72, p<2.2E-16, Pearson correlation). When stratified by decade, the distributions of the difference in parental AI overlap each other and the correlation in inferred parental AI global ancestry ranged from 0.65 to 0.74 (*Figure 3—figure supplement 4*), but were not statistically different from each other. This shows a consistent pattern of strong parental ancestry correlations among Mexican Americans over different generations. This signature of assortative mating is not due to recent parental relatedness, because there is no trend in long ROH with birth year (and an overall low rate of long ROH among Mexican Americans).

## Population genetic factors affecting changes in ancestry proportions over time

We developed a Moran model (*Moran, 1958*) style simulator to evaluate how migration, assortative mating, population growth, and variance in reproduction affect ancestry proportions over the time-scale shown in *Figure 2* (for details, see Materials and methods). Briefly, a Moran model is a forward simulation approach whereby each iteration, a single individual is replaced by another individual through a process of choosing parents. For a population of size $N$ individuals, it takes $N$ steps to simulate a single generation, and as such the Moran model is commonly used to represent overlapping generations.

In our simulations, the initial mean AI ancestry proportion was set at 0.42, and two generations were simulated (assuming ~26 years/generation) (*Moorjani et al., 2016*). Each iteration incorporated population growth, assortative mating, ancestry-based fecundity differences and migration. Simulating a standard neutral model of random mating and constant population size showed no change in ancestry proportions over time (*Figure 3—figure supplement 5*).

Population growth affects diversity by increasing number and proportion of variants that are rare, and decreasing the rate of genetic drift. While population growth can intensify the strength of natural selection (causing deleterious alleles to decrease in frequency and adaptive alleles to increase in frequency), population growth does not cause systematic changes in the frequency of segregating neutral alleles. Similarly, including population growth did not affect the mean ancestry proportion in a population (*Figure 3—figure supplement 6*).

In our simulations, we specified the assortative mating parameter to range from 0 (random mating) to 1 (parents are chosen as nearest neighbors when sorted by ancestry proportions). Ancestry-based assortative mating can lead to increased ROH and decreased genetic diversity (see *Figure 3*), but because mating occurs from individuals proportionally across the ancestry spectrum, ancestry-based assortative mating does not induce any changes in mean ancestry proportions in the population (albeit with slight increase in variance in ancestry proportions over time, *Figure 3—figure supplement 7*). Note that $AM = 0.75$ results in a correlation in parental ancestry proportions similar to our observed data.

There are many social and cultural properties that result in variance in fecundity within and between populations. Some of these factors may be correlated with genomic ancestry proportions.

We tested whether ancestry-based fecundity differences could induce changes in mean ancestry proportions, and how strong the fecundity differences had to be to induce an effect similar to what we see in the data. To simulate this process, we sampled individuals to reproduce based on their ancestry proportion using a $Beta\left(1, \frac{1}{(1+FAI)}\right)$ distribution, where $FAI = 0$ induces a uniform distribution (i.e. no ancestry-based fecundity differences) and $FAI = 1$ induces a strong ancestry-based fecundity difference (*Figure 3—figure supplement 8A*). Ancestry-based fecundity differences can induce systematic changes in ancestry proportions in the population (*Figure 3—figure supplement 8B–E*), but we are unaware of estimates of this effect in Mexican Americans. Further, ancestry-based assortative mating can magnify the effects of ancestry-based fecundity differences (*Figure 3—figure supplement 8F–I*). The joint effects of strong ancestry-related assortative mating ($AM = 0.75$) and fecundity differences ($FAI = 0.8$) results in a change in ancestry proportions over time similar to our observed data (*Figure 3—figure supplement 8I*).

While migration cannot explain all the changes in ancestry proportions we report, it is clearly a contributor. To model migration, we specified two parameters $mAI$: a parameter affecting migrant Amerindigenous ancestry proportions (*Figure 3—figure supplement 9*), and $M$: the probability that a new individual is a migrant. We simulated the joint effects of these parameters (*Figure 3—figure supplement 10*) and added the effects of ancestry-related assortative mating ($AM = 0.75$) and increasing degrees of ancestry-related differences in fecundity (*Figure 3—figure supplements 11–14*: FAI={0.1, 0.2, 0.4, 0.8}, respectively). We find a large number of parameter combinations that are consistent with our observed ancestry trends in Mexican Americans.

## Genetic association of global AI ancestry with biomedical traits

We have shown that genetic variation patterns changed over time in the Mexican American population, with AI ancestry increasing over a short period of time (combined with decreased genetic diversity and increased short and medium length ROH within AI ancestry tracts). These features may have implications for the genetic architecture of complex traits within Mexican Americans, a topic that is understudied and poorly understood. To further our understanding of the genetic architecture of complex traits in Mexican Americans, we investigated the relationship between AI ancestry and 69 biomedical phenotypes (while controlling for several environmental and other factors; see Materials and methods). As illustrated in *Figure 4A*, 18 of these traits (26%) are significantly associated with AI ancestry (Bonferroni correction p<6.6E-5) after adjusting for several factors including birth year, center, gender, sampling weight, educational attainment, US-born status, and number of US-born parents. While this suggests that genetic ancestry has an effect on several traits, other unmodeled socio-economic variables that are correlated with AI ancestry may also be contributing to these patterns (though AI ancestry has among the strongest effects on a range of biomedical traits, comparable to the effects of gender; *Figure 4—figure supplement 1*). Regardless, these findings highlight the need for increased investigation into the role of AI genetic ancestry in admixed populations such as Mexican Americans.

## Assessing the genetic contribution of AI ancestry to height

Among the traits we tested for association with global AI ancestry, height had the strongest effect. Further, our regression model indicated that height also had a strong positive relationship with birth year (*Supplementary file 3*). Globally, populations have grown taller over time due to a variety of non-genetic, environmental factors (*NCD Risk Factor Collaboration (NCD-RisC), 2016*). We find a similar trend in the HCHS/SOL Mexican Americans ($\beta$=0.096, 95% CI:0.077–0.114; p=5.95E-23) (*Figure 4B* and *Supplementary file 4*). Indeed, when we stratified individuals by quartiles of global AI ancestry, we see that all quartiles have increased in height by a similar amount over the period investigated (though individuals with lower AI ancestry were taller on average). The rates of change in height between AI quartiles were all positive and significant (p<5e-6). The largest was for the quartile with the highest AI ancestry, but the rates did not change monotonically with respect to AI ancestry across quartiles. The estimates for the quartiles with their 95% CIs are: $\beta$=0.135 (CI:0.097–0.173) for AI >0.58; $\beta$=0.124 (CI:0.089–0.160) for 0.46 <= AI <= 0.58; $\beta$=0.083 (CI:0.047–0.119) for 0.37 <= AI <= 0.46; and $\beta$=0.113 (CI:0.074–0.151) for AI <0.37 (*Supplementary file 4*).

Height is one of the most highly studied complex traits, with GWAS sample sizes numbering in the hundreds of thousands (*Yengo et al., 2018*). Results for many of these studies have been made

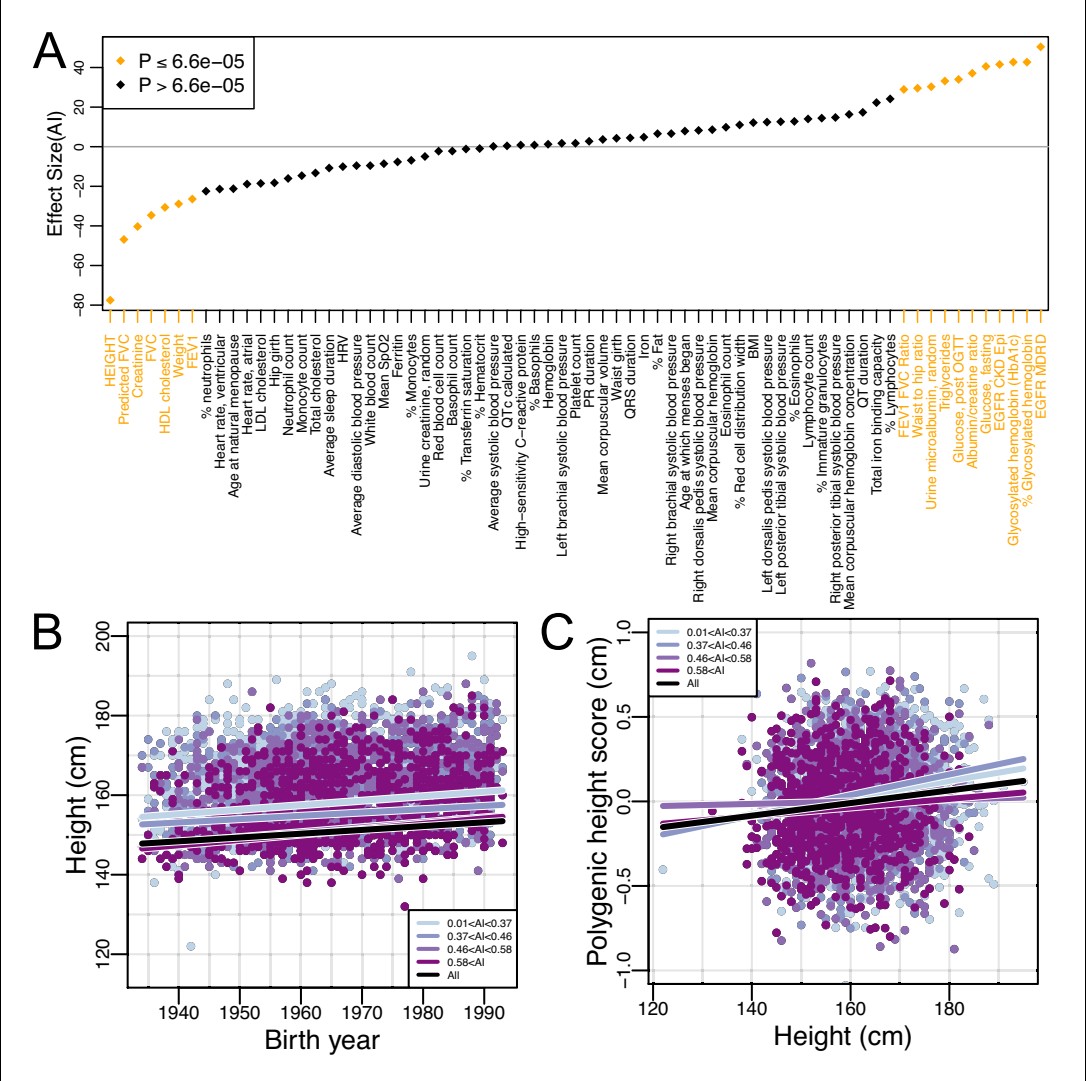

**Figure 4.** Global Amerindigenous ancestry and biomedical traits in HCHS/SOL Mexican Americans. (**A**) The effect size of global AI ancestry on each of 69 quantile normalized traits (see Materials and methods) while controlling for birth year, center, gender, sampling weight, educational attainment, US-born status, and number of US-born parents. (**B**–**C**) The relationship between (**B**) Birth year and height and (**C**) Height and polygenic height score (PHS). The black line indicates the fitted linear model for all individuals. Each color represents a different quartile of Amerindigenous global ancestry. Polygenic height scores were assessed utilizing UKBB summary statistics for 1,078 SNPs.

The online version of this article includes the following figure supplement(s) for figure 4:

**Figure supplement 1.** Distribution of variable effects associated with quantile normalized traits.

**Figure supplement 2.** Comparison of allele frequencies used in polygenic height score calculations.

**Figure supplement 3.** Polygenic height scores over time.

**Figure supplement 4.** Correlation of 69 p-values for Amerindigenous effect sizes of untransformed vs quantile normalized traits.

readily available on public databases as summary association statistics that can be leveraged to build genetic predictions through polygenic risk scores (PRS) (**Pasaniuc and Price, 2017**). In Europeans, PRS have been shown to have great predictive power for several traits, including breast cancer, prostate cancer, and type 1 diabetes (**Maas et al., 2016**; **Sharp et al., 2019**; **Maas et al., 2016**; **Schumacher et al., 2018**). PRS are most effective in populations of European descent as GWAS studies have been primarily performed in these populations (**Bustamante et al., 2011**;

*Martin et al., 2019*; *Popejoy and Fullerton, 2016*) and are expected to be biased when applied to other populations due to differences in the genetic architecture of traits across diverse populations (*Martin et al., 2017*). Since Mexican Americans have some fraction of European ancestry, we sought to determine whether PRS calculated utilizing GWAS summary statistics from European populations could still provide useful insight.

To evaluate the effectiveness of a PRS for height calculated based on 1078 genome-wide SNPs selected from the UKBB GWAS of height (i.e. the polygenic height score, or PHS, see Materials and methods), we first tested whether there was an association between the observed height and the predicted height estimates while controlling for sampling weight, gender, recruitment center, educational attainment, US-born status and number of US-born parents (see Materials and methods). Allele frequencies for these SNPs between the 1000 Genomes Americas Superpopulation and UKBB showed good concordance (*Figure 4—figure supplement 2*, r = 0.93, 95% CI:0.92–0.94, p<2.2E-16, Pearson correlation). We identified a significant association between observed height and predicted height for the population as a whole ($\beta$=0.004, 95% CI: 0.0023–0.005; p=9.91E-8; *Figure 4C*, *Supplementary file 5*). However, when we stratified by quartiles of AI global ancestry, the association only remained for the individuals in the lower two quartiles of global AI ancestry proportions (AI < 0.37: $\beta$=0.005, 95% CI:0.0022–0.0076; p=4.42E-4 and 0.36 < AI < 0.46: $\beta$=0.006, 95% CI: 0.0032–0.009; p=4.38E-5, *Supplementary file 5*). The association between predicted height and observed height was no longer significant for individuals in the upper two quartiles of global AI ancestry proportions (0.46 < AI < 0.58: $\beta$=0.0007, p=0.6 and 0.58 < AI: $\beta$=0.003, p=0.08, *Supplementary file 5*).

As we found global AI ancestry to be increasing over time (and there is a strong association between observed height and both AI as well as birth year), we hypothesized that there would be a change in PHS over time as well. However, we did not find a significant effect of birth year on PHS (*Figure 4—figure supplement 3*; p=0.14) even when we stratified by the quartiles of global AI ancestry.

## Discussion

The United States is a dynamic, rapidly changing population, and this will continue to occur as the population size grows (*Colby and Ortman, 2015*). Hispanics/Latinos are the largest and fastest growing minority group, and are projected to comprise ~29% of the US population by 2060. They are a genetically and phenotypically diverse population as a result of extensive admixture between Amerindigenous populations and immigrants from multiple geographic locations around the world. In this study, we identified additional population substructure complexities that may contribute to phenotypic variation within Hispanics/Latinos.

Specifically, we demonstrated how the admixture composition of Mexican Americans have changed over time, resulting in an increase of ~20% Amerindigenous ancestry on average over the 50 year period studied. This change in ancestry is equivalent to a mean increase in Amerindigenous ancestry of ~0.4% per year. While the effect sizes vary to some extent, we replicate the underlying pattern across multiple data stratifications (two metropolitan cities, US-born and non-US-born) and also replicate this feature in an independent cohort of Mexican Americans. Further, we find that a similar trend holds across multiple self-identified Hispanic/Latino populations in the US (and is statistically significant in Central Americans). This effect does not appear to have a simple explanation: we do not see any statistically significant increases in Amerindigenous ancestry at individual loci, we do not see more than a negligible degree of population differentiation over time, and this increase cannot be entirely explained by very recent migration based on our analyses of non-US-born individuals.

What could be driving the increased Amerindigenous ancestry in Mexican Americans? We hypothesize that several population, cultural, and environmental factors operating in unison have altered the genetic architecture of Mexican Americans. First, we identify strong ancestry-based assortative mating. However, while assortative mating could explain the increased ROH and decreased genetic diversity we inferred over time, ancestry-based assortative mating alone should not result in mean changes in global ancestry proportions (since a proportional number of offspring should derive from high- versus low-Amerindigenous ancestry parents, see simulations in *Figure 3—figure supplement 7*). Second, we do infer a subtle increase in Amerindigenous ancestry among individuals who migrated to the US more recently than individuals who migrated earlier.

Independent analyses have shown that migration from Mexico to the US has shifted over the years from states with less Amerindigenous ancestry to states with higher Amerindigenous ancestry (*Moreno-Estrada et al., 2014*; *Terrazas, 2010*). However, these subtle shifts in recent migration cannot fully explain the changes in Amerindigenous ancestry we infer, and taking them into account in our statistical model did not change the effect size that birth year has on Amerindigenous ancestry over time. Third, from US census data, we know that Hispanic/Latino is the fastest growing ethnicity (with Mexican Americans constituting the plurality). However, similar to assortative mating, population size changes alone should not drive mean changes in global ancestry proportions (*Figure 3—figure supplement 6*). While none of these factors alone can adequately explain the temporal dynamics of Amerindigenous ancestry we have observed, simulations of the joint effects of all of these factors operating in unison can indeed drive substantial changes in global ancestry patterns (*Figure 3—figure supplements 5–14*). However, more research is necessary to understand which parameters are consistent with the continuum of Mexican American populations across the US.

Regardless of the underlying mechanisms driving increased Amerindigenous ancestry in Mexican Americans, this additional source of temporal substructure within this population has substantial consequences for phenotypic variation in biomedical traits. We identify several biomedical traits that are associated with Amerindigenous ancestry, with effects comparable to the high effects of gender, and show that in the case of height, there are both ancestry and temporal effects. While we do see differences in mean height based on percentage of AI ancestry, height increases over time in all groups at similar rates. Individuals with lower percentages of AI ancestry were taller on average than individuals with higher AI ancestry pointing to the role of AI ancestry on the trait. Further study is necessary to understand whether other biomedical traits are also changing over time as a result of the change in genomic ancestry proportions, and the degree to which other socio-economic factors independently drive both ancestry patterns as well as biomedical traits.

In our study, we bring specific attention to the biases that continue to exist with using European GWAS summary statistics to calculate polygenic risk scores in admixed populations such as Mexican Americans that are comprised of European, Amerindigenous, and African genetic ancestries. In particular, in the case of height, we found that the polygenic height score (PHS) correlated with observed height only in the subset of individuals with the lowest levels of Amerindigenous ancestry (i.e. the subset of individuals with highest European ancestry). As the population dynamics of the US continue to change, it is imperative that we study diverse populations, or we risk exacerbating the health disparities that currently exist. To date, population-based medical genomics research (and its subsequent benefits) have been disproportionately focused on populations of European ancestry. In order to improve the design and implementation of medical genetics studies for the ethnically diverse U.S. population, we need detailed insights into the population history of diverse U.S. populations. This includes characterizing the admixture dynamics of Hispanic/Latino populations, as well as the evolutionary forces that shaped patterns of genetic variation of the ancestral populations that contributed to modern day Hispanic/Latino populations.

The genetic variation of the Hispanic community in the United States belies categorization under a single label (*Conomos et al., 2016*). The events that have shaped and continue to shape this genetic diversity are complex, numerous, and nuanced, and the social history of such a diverse population is intrinsic to any genetic study. Mexico's society was largely defined by an established social caste system based on ancestry, which disappeared after Mexico's independence in 1821 (*Lisker et al., 1990*). Even so, social inequalities persist today with skin color having a significant effect on wealth and education (*Martinez, 2017*). A multitude of factors within and outside Mexico — whether related to trade, immigration policies, or armed conflicts — acted to influence who immigrated to the United States, and the impact of each of these fluctuates over time (*Contreras, 2014*; *Verea and Verea, 2014*; *Fernández-Kelly and Massey, 2007*). These changes shift the demographics of immigration, which is inherently related to the genetic ancestry of the population.

Consequently, this shapes the genetic architecture of complex traits. Diverse populations are at risk not only from underrepresentation in research, but because of poor understanding of the temporal and spatial dynamics at play in genetic variation. The promise of equitable precision medicine — one of the ultimate goals of medical genomics — cannot be kept without understanding this interplay. Health disparities in the United States are fed by structural inequalities. For example, studies that use modern Artificial Intelligence techniques have already been shown to inflate existing

disparities between Black Americans and White Americans (*Obermeyer et al., 2019*). Such biases, whether from algorithms, study designs, or misunderstandings of subtleties in data, feed into the larger systemic pressures faced by minority populations in the United States.

While we have shown a dramatic shift in ancestry proportions in US Hispanic/Latinos, one of the caveats of this study is that the HCHS/SOL cohort is not representative of all US Hispanics/Latinos. HCHS/SOL participants were recruited at four primary centers: Bronx, Chicago, Miami, and San Diego. There may be additional genetic diversity that has not been captured by this dataset and trends exhibited in this dataset may not translate to Hispanic/Latino populations living in other regions of the US (though the temporal increase in Amerindigenous ancestry was replicated in an independent sample of Mexican Americans). Further, we have only assembled a reference panel with limited numbers of individuals with various Amerindigenous, European, and African ancestry. With better population genetic modeling and a deeper understanding of the social and historical aspects of Hispanic/Latino populations, we will be able to improve our understanding of the genetic and phenotypic diversity across these populations, and subsequently improve our ability to understand genetic contributions to complex traits and disease. These insights will lead to optimization of population sampling for the design of future medical genetic studies, the identification of disease risk variants, and ultimately, precision medicine for all.

## Materials and methods

### Study dataset and initial quality control

The HCHS/SOL study is a community-based cohort study of self-identified Hispanic/Latino individuals from four US metropolitan areas with the general goal of identifying risk and protective factors for various medical conditions including cardiovascular disease, diabetes, pulmonary disease, and sleep disorders (*Sorlie et al., 2010*). The sample survey for design for HCHS/SOL has been described previously (*LaVange et al., 2010*). Briefly, census block groups were selected in defined communities near each of the four recruitment centers, and households were sampled within census block groups. Households with Hispanic/Latino surnames and individuals as well as residents over 45 years old were oversampled in order to increase representation of the Hispanic/Latino target population and achieve a uniform age distribution. Sampling weights were calculated for each individual to reflect the probability of sampling (*Conomos et al., 2016*). 12,434 participants with birth year estimates between 1934–1993 who self-identified as being of Cuban, Dominican, Puerto Rican, Mexican, Central American, or South American background consented to genetics studies and posting of their genetic and phenotype data on the publicly available Database of Genotypes and Phenotypes (dbGaP) through Study Accession phs000810.v1.p1. Samples were genotyped on an Illumina custom array, SoL HCHS Custom 15041502 array (annotation B3, genome build 37), consisting of the Illumina Omni 2.5M array and 148,353 custom single nucleotide polymorphisms (SNPs) (*Conomos et al., 2016*). Data posted to dbGaP had passed initial sample quality control filters, including removing samples with differences in reported vs. genetic sex, call rates > 95%, and evidence for sample contamination (e.g. heterozygosity and sample call rates). For initial SNP quality control, we filtered out SNPs that were monomorphic, positional duplicates, or Illumina technical failures, as well as SNPs that had cluster separation <= 0.3, call rate <= 2%,>2 discordant calls in 291 duplicate samples,>3 Mendelian errors in parent-offspring pairs/trios, Hardy-Weinberg Equilibrium combined p-value$<10^{-5}$, and sex differences in allele frequency $\geq$0.2. Our filtering resulted in 1,763,935 genotyped SNPs with minor allele frequency (MAF) >0.01.

Additional sample quality control performed in the HCHS/SOL dataset included filtering out samples with (1) large chromosomal anomalies, (2) substantial Asian ancestry as previously identified in HCHS/SOL (*Conomos et al., 2016*) and (3) individuals with up to third degree genetic relatedness in the dataset as inferred by REAP (*Thornton et al., 2012*). For genetic relatedness filtering, individuals from pairs were kept to maximize representation of the birth year distribution, which resulted in 10,268 unrelated remaining individuals.

From the original HCHS/SOL analysis, individuals were classified into genetic-analysis groups, similar to self-identified background groups in that they share cultural and environmental characteristics, but are also more genetically homogenous (*Conomos et al., 2016*).

Birth year for all individuals was estimated by subtracting the difference between date of first clinic visit for the baseline examination (*Sorlie et al., 2010*) and age. Year of arrival was estimated by subtracting the difference between date of first clinic visit for the baseline examination and years in the US.

## Global, local, and parental ancestry inference

All ancestry analyses were restricted to the 211,152 autosomal SNP markers that overlapped between the study and reference panel genotyping array. For the HCHS/SOL dataset, global African, European, and Amerindigenous ancestries were inferred with ADMIXTURE (*Alexander et al., 2009*); in an unsupervised manner, with K = 3. Amerindigenous ancestry refers to estimates of Indigenous genetic ancestry from the Americas. For some analyses, HCHS/SOL individuals with greater than 95% of a single ancestry (e.g African, European, or Amerindigenous) were filtered out resulting in 9913 individuals: 1,099 Central American, 1,536 Cuban, 954 Dominican, 3,622 Mexican, 1,783 Puerto Rican, 652 South American and 267 'Other' individuals.

Ancestral tracts, known as 'local' ancestry, along the genome for all HCHS/SOL individuals were inferred using RFMix (*Maples et al., 2013*) and a three population reference panel, comprised of 315 individuals: 104 HapMap phase 3 CEU (European) and 107 YRI (African) individuals (*Altshuler et al., 2010*) and 112 Amerindigenous individuals from throughout Latin America (*Reich et al., 2012*). The reference panel was limited to individuals with 99% continental ancestry as inferred by unsupervised ADMIXTURE (*Alexander et al., 2009*). Prior to local ancestry inference, HCHS/SOL individuals were merged with the reference panels and then phased using SHAPEIT2 (*Delaneau et al., 2013*). For all HCHS/SOL Mexican American individuals, parental genomic ancestry was inferred with ANCESTOR (*Zou et al., 2015*) using the local ancestry estimates generated by RFMix.

Bootstrap analyses (*Figure 2B* and *Figure 2—figure supplement 3*) were performed by calculating relevant statistics based on repeated resampling of individuals with replacement. Bootstrap resampling results in an estimate of the variance of the statistics that we are calculating in our data, and allows us to assess the impact of outliers (who are only resampled in a subset of iterations).

## Uniform manifold approximation and projection (UMAP)

Principal components for HCHS/SOL and the reference panel were computed using smartPCA (*Patterson et al., 2006*). UMAP (version 0.3.8) was run using the Python script freely available at (https://github.com/diazale/gt-dimred; *Diaz-Papkovich, 2019*) with parameter specification set at 15 nearest neighbors and a minimum distance between points of 0.5.

For further analyses of HCHS/SOL population structure, a larger reference panel was assembled comprising of additional European and African populations from the Human Genome Diversity Project (HGDP) (*Rosenberg et al., 2002*; *Reich et al., 2012*) and 1000 Genomes Project (*Auton et al., 2015*). For the European reference panel, 24 Basque, 28 French, 12 Italian, 25 Russian, and 28 Sardinian individuals from HGDP and 90 GBR, 107 IBS, 99 FIN, and 107 TSI individuals from 1000 Genomes were included with the original 104 CEU individuals. For the African reference panel, 9 BantuKenya, 8 BantuSouthAfrica, 22 Mandenka, 26 Mozabite from HGDP and 99 ESN, 113 GWD, 97 LWK, and 82 MSL from 1000 Genomes were included with the original 107 YRI individuals. Combined with the 112 Amerindigenous and 10,268 HCHS/SOL individuals, the larger additional analyses comprised 11,567 individuals in total.

## Admixture mapping

Local ancestry estimates for 211,151 SNPs across the genome were used to perform admixture mapping in HCHS/SOL Mexican Americans to determine if younger individuals harbored excess Amerindigenous ancestry in certain regions of the genome. Admixture mapping was performed applying two different models: (1) a linear regression model with age as the dependent variable adjusting for global Amerindigenous ancestry, sampling weight and center and (2) a logistic regression model dividing the HCHS/SOL Mexican cohort in to an older vs younger generation with 1965 set as the dividing point while also adjusting for global Amerindigenous ancestry, sampling weight, and center. The threshold for genome-wide significance, $1.38 \times 10^{-4}$ was calculated using the empirical

autoregression framework with the package *coda* in R to estimate the total number of ancestral blocks (*Sobota et al., 2015*; *Plummer et al., 2012*).

### Tract lengths

The multiple regression model: $\log(f) = \beta_0 + \beta_1\ T + \beta_2\ A + \beta_3\ TA + \varepsilon$, where $f$ is a matrix containing the proportion of lengths of all ancestral tracts across the genome for all 3622 Mexican American individuals, $T$ the tract length bin and $A$ decade of birth year bin, was used to test for an effect of birth-decade on the proportion of Amerindigenous ancestral tract lengths. For assessment between the fraction of ancestry tracts in an individual's genome and birth year, long tract cutoffs were chosen based on tract separation between the birth year decades in *Figure 3B*.

### Diversity calculations

Subcontinental ancestry was assessed using the diversity measurements $\pi$ and $F_{ST}$. $\pi$ was calculated as the average number of pairwise genetic differences among all pairs of overlapping Amerindigenous ancestry tracts across individuals. $F_{ST}$ was calculated as:

$F_{ST} = (H_T - H_S)/H_T$ where $H_T$ is the average heterozygosity when all individuals are pooled across decades and $H_S$ is the average heterozygosity within each decade of individuals.

### Inference of runs of homozygosity

ROH were called using the program GARLIC v1.1.4 (*Szpiech et al., 2017*) on 211,152 sites for the Mexican American individuals. An analysis window size of 50 SNPs and an overlap fraction of 0.25 were both chosen using GARLIC's rule of thumb parameter estimation. GARLIC chose a LOD score cutoff of 0. Using a three-component Gaussian mixture, GARLIC determined class A/B (short/ medium) and class B/C (medium/long) size boundaries as 845,097 bp and 2,501,750 bp, respectively.

### Simulating ancestry proportions over time

Our Moran model simulator includes population growth (exponential), migration (with adjustable levels of migration and ancestry patterns), ancestry-based assortative mating, and ancestry-based variability in fecundity (see https://github.com/mlspear09/hchs-sol; *Spear, 2020*). Our simulations are modeled after the data shown in *Figure 2*. Initial ancestry proportion in the population was set to 0.42. Previous estimates of the generation time in humans has resulted in an estimate of ~26–30 years per generation (*Moorjani et al., 2016*). As such, the data analyzed correspond to ~2 human generations. We therefore begin our simulations with a random sample of ancestry proportions with mean 0.42, and simulate two generations (corresponding to *2N* steps in our simulator). In all simulations, we start with *N = 1000*. The general idea is to model population parameters (such as average ancestry proportions in the population), which is less sensitive to the actual population size used.

### Imputation

Imputation for HCHS/SOL was performed locally using IMPUTE2 (*Howie et al., 2009*) with the 1000 Genomes Project Phase three haplotypes (*Auton et al., 2015*) used as a reference panel. After filtering on an info score cutoff of 0.3, this resulted in 33,041,084 SNPs.

### Analyzing biomedical traits

We analyzed a total of 69 biomedical traits contained in the HCHS/SOL phenotypic dataset. We used a multiple linear regression model to analyze the effects of global AI ancestry on each trait while controlling for birth year (a proxy for age), center, gender, sampling weight, educational attainment, US-born status, and number of US-born parents. In *Figure 4A*, we show the effect size ($\beta$) for AI ancestry after quantile normalizing each trait (*Bolstad et al., 2003*; *Qiu et al., 2013*). Quantile normalization forces each phenotype to be rank-transformed to a Standard Normal distribution. While quantile normalization is a common approach to transforming data to conform to the Normal distribution assumption inherent in linear regression (and provides the benefit of effect sizes that are readily comparable across traits), this procedure can result in a modest reduction in statistical power compared to untransformed data (*Qiu et al., 2013*). We find that the p-values for the AI effect sizes are highly correlated when phenotypes are untransformed vs quantile normalized (Spearman

$\rho$=0.943; p<2.2e-16) (*Figure 4—figure supplement 4*) with no statistical evidence for a difference in their distribution (Mann-Whitney U test p=0.912).

## Polygenic risk score calculations

Polygenic risk scores for height were calculated using the publicly available UK Biobank (UKBB) GWAS Round 2 Summary Statistics retrieved from http://www.nealelab.is/uk-biobank. Briefly, for sample quality control, sample inclusion was limited to unrelated samples who passed the sex chromosome aneuploidy filter. British ancestry was determined using the 1st 6 PCs; individuals more than seven standard deviations away from the 1st 6 PCs were excluded. Further filtering included limiting to self -reported 'white-British' / 'Irish' / 'White' resulting in a QCed sample count of 361,194 individuals as described in (https://github.com/Nealelab/UK_Biobank_GWAS#imputed-v3-sample-qc; *Neale Lab, 2018*). An imputation panel of ~90 million SNPs from HRC, UK10K and 1 KG were used to impute genotypes. 13.7 million autosomal and X-chromosome SNPs passed quality control thresholds including Info score >0.8, MAF >0.0001, and HWE p-value>1e-10. For the phenotype, a linear regression model in Hail was run for all individuals (both sexes) adjusting by the first 20 PCs + sex + age + age$^2$ + (sex*age) + (sex*age)$^2$. For height, there was complete phenotype information for 360,388 individuals.

Risk scores were calculated by extracting the overlapping genome-wide significant hits initially discovered in the UKBB GWASs of height and selecting SNPs with the lowest p-value in each 1 Mb window across the genome. Prior to extraction there were a total of 227,794 genome-wide significant SNPs initially discovered in the UKBB GWAS of height. For height this resulted in a dataset of 1078 overlapping SNPs for the PRS calculation that were present in our dataset of genotyped and imputed SNPs.

## Health and retirement Study (HRS)

For replication, we used genotype data from 705 self-identified Mexican Americans from the Health and Retirement Study (HRS) (*Fisher and Ryan, 2018*), genotyped on the Illumina Human Omni 2.5M platform. HRS data was made available under IRB Study No. A11-E91-13B - The apportionment of genetic diversity within the United States. Estimated global ancestry proportions for the Mexican American population in the HRS were calculated as in *Baharian et al., 2016*, which used an alternative reference panel and alternative ancestry inference approach. Briefly, RFMix was used to infer local ancestry estimates across the genome utilizing CHS, YRI, and CEU individuals from the 1000 Genomes Project as reference populations for Amerindigenous/Asian, African, and European ancestries, respectively. Global ancestry estimates were calculated using the summed RFMix calls.

## Statistical analyses and plots

Statistical analyses and plot generation were performed within Rstudio using Version 1.1.463 and R version 3.5.3. ternary and ggridges/ggplot2 packages were used to create the simplex and ridgeline plots.

For each of the HCHS/SOL populations, we evaluated differences in global ancestry estimates over time while accounting for the sampling method (referred to as 'sampling weight', see Materials and methods) used for the design of the HCHS/SOL study.

To test for differences in each ancestry over time for each HCHS/SOL population, we ran a linear regression model of Ancestry = $\beta_0$ +$\beta_1$ BY +$\beta_2$ $SW$ + $\varepsilon$, where BY = birth year and SW = log(sampling weight). Within the Mexican Americans, we ran this model stratified by gender, the recruitment centers Chicago and San Diego, born in the US versus outside the US and education attainment. For recruitment centers, data stratification was limited to Chicago and San Diego as sample size for the Bronx and Miami was limited: 124 and 25 individuals, respectively. Education attainment was categorized as either less than a high school diploma or equivalent degree (<HS), equal to a high school diploma or equivalent degree (=HS), or post-secondary education (>HS).

To test differences in mean Amerindigenous ancestry by group, we ran t-tests. The data were split and compared by gender, recruitment center, born in the US versus outside the US, and educational attainment levels.

For the height and polygenic height score analyses, 3604 Mexicans were included based on complete information for height, gender, recruitment center, sampling weight, education attainment, born in the US versus outside the US, and number of US-born parents.

## Acknowledgements

We thank many colleagues who commented on our preprint prior to submission, particularly Reed Cartwright for suggestions on terminology. MLS was supported through the National Human Genome Research Institute (NHGRI) of the National Institutes of Health (NIH) under Award Number F31HG010104. ADP and SG were supported, in part, thanks to funding from the Canada Research Chairs program and CIHR grant MOP-136855. EZ was supported, in part, by NIH grants R0184545 and K24CA169004. RDH was supported, in part, by NHGRI grant R01HG007644 and the Canadian Research Chairs program.

## Additional information

### Funding

| Funder | Grant reference number | Author |
| --- | --- | --- |
| National Institutes of Health | R01HG007644 | Ryan D Hernandez |
| National Institutes of Health | F31HG010104 | Melissa L Spear |
| Canadian Institutes of Health Research | MOP-136855 | Simon Gravel |
| National Cancer Institute | R01184545 | Elad Ziv |
| National Cancer Institute | K24169004 | Elad Ziv |

The funders had no role in study design, data collection and interpretation, or the decision to submit the work for publication.

### Author contributions

Melissa L Spear, Conceptualization, Resources, Data curation, Software, Formal analysis, Funding acquisition, Investigation, Visualization, Methodology, Writing - original draft; Alex Diaz-Papkovich, Resources, Formal analysis, Visualization, Writing - review and editing; Elad Ziv, Conceptualization, Resources, Data curation, Formal analysis, Visualization, Methodology, Writing - original draft; Joseph M Yracheta, Formal analysis, Visualization, Writing - review and editing; Simon Gravel, Dara G Torgerson, Resources, Supervision, Writing - review and editing; Ryan D Hernandez, Conceptualization, Resources, Supervision, Methodology, Writing - review and editing

### Author ORCIDs

Melissa L Spear https://orcid.org/0000-0002-3252-8411
Alex Diaz-Papkovich https://orcid.org/0000-0002-2867-5494
Elad Ziv https://orcid.org/0000-0002-2324-2884
Joseph M Yracheta https://orcid.org/0000-0002-0691-8504
Simon Gravel https://orcid.org/0000-0002-9183-964X
Ryan D Hernandez https://orcid.org/0000-0001-5249-504X

### Decision letter and Author response

Decision letter https://doi.org/10.7554/eLife.56029.sa1
Author response https://doi.org/10.7554/eLife.56029.sa2

# Additional files

## Supplementary files

• Supplementary file 1. Association of global ancestries and birth year for all HCHS/SOL individuals. For each population, we tested for an association between global ancestry and birth year while accounting for the sampling design. AI, AFR, and EUR refer to Amerindigenous, African, and European ancestry respectively. The significance threshold was set at 0.003 using Bonferroni correction for multiple testing (0.05/18).

• Supplementary file 2. Frequency table of 3622 HCHS/SOL Mexican Americans stratified by recruitment region, US-born vs non-US-born status, gender and educational attainment. Recruitment was performed at four regions: Bronx, Chicago, Miami and San Diego. Education attainment was categorized as either less than a high school diploma or equivalent degree (<HS), equal to a high school diploma or equivalent degree (=HS), or post-secondary education (>HS).

• Supplementary file 3. Association of quantitative traits and Amerindigenous ancestry in HCHS/SOL Mexican Americans. Each trait as a function of AI ancestry adjusted by birth year, center, gender, sampling weight, educational attainment, US-born status, and number of US-born parents. Results are shown for both the raw data and quantile normalized data.

• Supplementary file 4. Height over time. Height (cm) as a function of birth year adjusting by center, gender, sampling weight, educational attainment, US-born status, and number of US-born parents for 3604 Mexican Americans stratified by the quartiles of global Amerindigenous ancestry (AI).

• Supplementary file 5. Predicted height vs. observed height. Predicted height (cm) as a function of observed height (cm) adjusting by center, gender, sampling weight, educational attainment, US-born status, and number of US-born parents for 3604 Mexican Americans stratified by Amerindigenous ancestry (AI).

• Transparent reporting form

## Data availability

All data used in this manuscript were downloaded from publicly available sources (dbGap). No new data were created.

The following previously published datasets were used:

| Author(s) | Year | Dataset title | Dataset URL | Database and Identifier |
|---|---|---|---|---|
| Conomos MP, Laurie CA, Stilp AM, Gogarten SM, McHugh CP, Nelson SC | 2016 | Genetic Diversity and Association Studies in US Hispanic/Latino Populations: Applications in the Hispanic Community Health Study/Study of Latinos | https://biolincc.nhlbi.nih.gov/studies/hchssol/ | phs000810.v1.p1, phs000810.v1.p1 |
| Fisher GG, Ryan LH | 2018 | Overview of the Health and Retirement Study and Introduction to the Special Issue | https://hrs.isr.umich.edu/data-products/access-to-public-data | A11-E91-13B, A11-E91-13B |

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
