## [Decision Letter]

**Acceptance summary:**

This article studies genetic and complex trait variation in individuals in the United States with origins in Mexico. The authors find a pattern of increasing amerindigenous ancestry with birth year, which they investigate using simulations and attribute to the likely combination of several cultural and historical factors. They find amerindigenous ancestry to be associated with trait variation for several complex traits, and highlight the importance of further work in medical and population genomics across human diversity.

**Decision letter after peer review:**

Thank you for submitting your article "Recent fluctuations in Mexican American genomes have altered the genetic architecture of biomedical traits" for consideration by *eLife*. Your article has been reviewed by three peer reviewers, including Mashaal Sohail as the Reviewing Editor and Reviewer #1, and the evaluation has been overseen by Patricia Wittkopp as the Senior Editor. The following individual involved in review of your submission had agreed to reveal their identity: Genevieve L Wojcik.

The reviewers have discussed the reviews with one another and the Reviewing Editor has drafted this decision to help you prepare a revised submission.

The editors have judged that your manuscript is of interest, but as described below that some conclusions and analyses need to revised as presented in light of our comments before it is published. We would like to draw your attention to changes in our revision policy that we have made in response to COVID-19 (https://elifesciences.org/articles/57162). First, because many researchers have temporarily lost access to the labs, we will give authors as much time as they need to submit revised manuscripts. We are also offering, if you choose, to post the manuscript to bioRxiv (if it is not already there) along with this decision letter and a formal designation that the manuscript is “in revision at *eLife*”. Please let us know if you would like to pursue this option. (If your work is more suitable for medRxiv, you will need to post the preprint yourself, as the mechanisms for us to do so are still in development.)

Summary:

This paper examines population structure in a Hispanic/Latino (H/L) cohort, first highlighting genetic diversity and fine-scale population substructure via UMAP. The paper highlights an interesting temporal aspect to population substructure in H/L groups, demonstrating increasing proportions of Amerindigenous ancestry, particularly in Mexican Americans, over time. They also provide other analyses that may help interpret or follow from this observation, involving genetic diversity, assortative mating and runs of homozygosity. They show a correlation between amerindigenous ancestry and complex traits, and show that the behavior of UKB height GWAS polygenic scores in Mexican-Americans depends on the proportion of Amerindigenous ancestry.

Essential revisions:

This paper's major finding is an observation of an increase in Amerindigenous ancestry in Mexican Americans in the 1940-1990 period. The manuscript is interesting and in theory suitable for publication in *eLife*, but after a number of points are addressed. First, the authors need to articulate clearly the factors that may have caused this primary observation, and what may be the most likely explanations outlined below. Second, they need to address that the primary explanation for the ROH increase is likely the amerindigenous ancestry increase, and in that sense, determine and clearly articulate the place of the assortative mating observation in the manuscript. Lastly, they need to clearly admit in the manuscript the importance of the un-modelled socio-economic variable in the correlation between amerindigenous ancestry and complex traits, and only present this analysis after controlling for appropriate covariates. It is a time when we are finally seeing some population genetic studies of understudied populations as they relate to complex trait variation (to which this manuscript can be an important contribution), and so the bar to be as rigorous as possible in considering alternative explanations and model all possible covariates should be set extremely high.

1) The main finding is the increase in amerindigenous ancestry in the 1940 – 1990 period in Mexican-Americans in the United States.

a) The authors state: "In our non-US born individuals (N=2987), we evaluated differences in ancestry estimates over time while accounting for years in the US and sampling weight and identified a significant effect of years in the US (𝛽=-0.0009; P=0.0006; SE=0.0003)." If I understand correctly, this regression is amerindigenous ancestry against time and other covariates. It would be helpful if the authors add to this sentence something along the lines of, "implying that individuals who arrived earlier to the US from Mexico had more European ancestry."

b) Given the above analysis, and independent migration analyses (see, for example, https://www.migrationpolicy.org/article/mexican-immigrants-united-states-2), it seems that migration from Mexico to the US shifted over the years from states with less amerindigenous ancestry to states with higher amerindigenous ancestry (Chiapas, Oaxaca, Veracruz) in the South and South-east of the country. This seems a highly likely explanation for the pattern of increasing amerindigenous ancestry that they see, and should be stated as such in the manuscript. This seems especially likely given that the signal comes primarily from non-US born individuals, or US-born individuals with parents born outside the US.

c) The authors state that "It is possible that the increase in global Amerindigenous ancestry over time could be biased by changes in the specific subcontinental Amerindigenous ancestries over time (though such an effect is not visible in our UMAP analysis, Figure 1B)." – It is not clear what is meant by this sentence – please re-phrase and articulate more clearly. If it alludes to the difference in migration sources over time I mention above, I don't think their analyses of Fst and genetic diversity rule that explanation out.

d) Assortative mating (Figure 2D and Supplementary figure 9). This argument is puzzling because if there is assortative mating along indigenous ancestry as they suggest, then would this not mean that there is also assortative mating along the collinear European ancestry? If this is the case, why would amerindigenous ancestry be increasing in particular? The authors do state that assortative mating would not cause an increase in one ancestry. In that case, the paper overall does not provide an explanation for why the amerindigenous ancestry is actually increasing – is the migration sources explanation the most likely explanation? Along with that individuals with higher amerindigenous ancestry must be reproducing more? The likely explanations of the primary result should be made very clear for the reader.

e) The authors state: "and this increase cannot be entirely explained by very recent migration." What is the evidence backing this claim?

f) "sampling weight" is not actually defined in the Materials and methods. Can the authors clarify how this is defined and used to weight the major analysis?

g) Please describe the bootstrap resampling performed – is the bootstrapping performed over individuals or segments of the genome? Please justify the strategy picked. This should be described in the manuscript.

2) The pattern of ROH change over time observed.

a) What are the number of individuals in each decade? Please show this in the manuscript. If there are more individuals in later birth decades (as may be expected), you would see an increase in the ROH summed over each genome with time, simply because you are summing over more individuals at later time periods. It is not clear if the analyses for Figure 2C are normalized by the number of individuals in each decade – if not, this would be important to do, and only the normalized results should be reported.

b) The ROHsum increasing with time could simply be due to the amerindigenous ancestry increasing with time, as amerindigenous ancestry carries more short ROH segments than European ancestry (see for example Ceballos et al. Nature Review Genetics 2018). The authors should explicitly describe this, as this simplest explanation would not require assortative mating to be invoked either.

3) Correlation of amerindigenous ancestry and complex traits

Many of the traits studied would also be affected by socioeconomic status (for example, height, cholesterol). Do the authors have this variable available? If yes, it should be included in the multiple regression. If not, it should be clearly mentioned that they are not able to account for this likely important effect, leaving their estimates confounded by socio-economic differences that likely correlate with amerindigenous ancestry. For Figure 3, we don't think it is fair to show tau between only amerindigenous ancestry and traits as this analysis does not account for important covariates, and would like to see Supplementary file 3 instead to replace Figure 3 (in a figure form as the authors prefer) such that only the multiple regression effect sizes are reported in the manuscript that account for covariates.

Why do a first pass of this analysis without covariates included, and then re-run with covariates in the Bonferroni significant subset of traits only? (Given that there will be confounding between Amerindigenous ancestry and socioeconomic, environmental and other non-genetic factors, and especially age). Furthermore, looking in the Materials and methods section, we cannot seem to find the full description of how this analysis was performed i.e. what models were run, and how/if phenotypic measures used were cleaned and normalized etc. Please provide this.

[Editors' note: further revisions were suggested prior to acceptance, as described below.]

Thank you for the revised submission of your manuscript, now called "Recent shifts in the genomic ancestry of Mexican Americans may alter the genetic architecture of biomedical traits." for consideration by *eLife*.

The Reviewing Editor has drafted this decision to help you prepare a revised submission.

Summary:

The manuscript is significantly improved, and the addition of the new simulation analyses greatly help in the interpretation of the trends that they see. The authors have sufficiently addressed concerns about two of their main results: (1) Interpretation of ancestry change over time and (2) Interpretation of ROH change over time. I still have the following points I would ask them to consider regarding their third main result (3) the potential effects of ancestry and ancestry change over time on the genetic architecture of complex traits, and to make appropriate additions/revisions to their analyses and text before submitting a revised manuscript. Beyond that, I see that the new simulation results don't appear in the manuscript until the Discussion. I would consider them a result and would like to see the authors try to integrate the main simulation results in the Results section, when they report their empirical observed patterns.

The authors state in the manuscript that, "As illustrated in Figure 4A, 20 of these traits (29%) are significantly correlated after Bonferroni correction (P<0.000145), highlighting the need for increased investigation into the role of AI genetic ancestry and other unmodelled socio-economic variables in admixed populations such as Mexican Americans." As written, it understates the correlation between AI ancestry and unmodelled socio-economic variables in the United States which we know exists on the basis of historical and social science research. Given this, I would like to see the text revised to say that it is not clear what the correlation with AI ancestry implies, and while it could be reflecting genetic effects, it can also be reflecting socio-economic variables that are correlated with AI ancestry and that AI ancestry could be serving as a proxy for. The authors themselves show that AI ancestry is correlated with educational attainment levels which they state is a proxy for socio-economic status. I would like to see their model for testing the effect of ancestry on traits include as covariates: (1) educational attainment as a proxy for socioeconomic status, (2) whether they are US born or not, and whether their parents are US born or not, to help capture the effects of different environments they would have been born in, and that their parents would have created for them on various levels. I would also encourage the authors to make their model as rich as possible by adding other environmental variables they could obtain on their recruitment sites (altitude, latitude, longitude, population density, average obesity rates to name some that are likely relevant). Ideally, this kind of analysis would be done in a mixed model framework as well, correcting for the full genomic relationship matrix, and adding a random effect to account for unaccounted for environmental factors, but they should at least add covariates to their multiple regression framework that they have access to, or could access. They should also consider issues of collinearity as they may affect their estimates and study and report the Variance Inflation Factors (VIF) of the different variables. They should report in the manuscript, the results for the full model, giving coefficients and p-values for not only AI ancestry but also the other test variables, and should describe these in the results as well, and report and discuss the contribution of other test variables relative to AI ancestry as estimated from their model.

Further, the authors results show three observations that I would like to see described in the Results and discussed in the Discussion, as the implications are important. The authors observe an increase in height in Mexican Americans (at roughly the same rate in all amerindigenous ancestry stratifications, see note below) with birth year. First, they do not see any trend of the polygenic height score with birth year. This suggests that while the genetic predisposition of the trait remains the same, the trait has changed significantly due to non-genetic environmental factors. Second, even though amerindigenous ancestry is negatively correlated with height, and amerindigenous ancestry is increasing over time, the trait value/height increases rather than decreases over time. This also points to the effects of non-genetic factors playing an important role in values of the trait.

Lastly, if amerindigenous ancestry is negatively correlated with height due to genetic reasons, then shouldn't we expect to see the polygenic height score decrease with birth year, as amerindigenous ancestry increases with birth year? How do the authors interpret this meta pattern across their analyses, and what implications does it have for how temporal change in ancestry can alter the genetic architecture of traits? Overall, this could mean that (1) the correlation of amerindigenous ancestry with height is at least partly due to genetic reasons. Given that, while ancestry trends (AI ancestry increasing), combined with correlations of ancestry with traits, may make you predict one thing with respect to the genetic architecture of traits (height will decrease with birth year), the way heredity interacts with the environment through the randomness of development makes the trait move in the opposite direction than the model would predict. Or (2) the correlation of amerindigenous ancestry with height is fully picking a signal of height being lower in individuals with higher amerindigenous ancestry due to primarily environmental reasons, and therefore, as environment changes, the correlation is not meaningful for prediction. I would like to see the authors consider the above, and state these patterns as they stand across analyses, and discuss their implications for their overall thesis (as it relates to height and traits in general).

Note: They say in the study, "We find a similar trend in the HCHS/SOL Mexican Americans (Figure 4B). Indeed, when we stratified individuals by quartiles of global AI ancestry, we see that all quartiles have increased in height by a similar amount over the period investigated." Can the authors make this statement more specific in the manuscript – what are the rates of change in the different quartiles? Are they higher in quartiles with higher indigenous ancestry? Please integrate this into the Discussion above as well.

[Editors' note: further revisions were suggested prior to acceptance, as described below.]

Thank you for resubmitting your work entitled "Recent shifts in the genomic ancestry of Mexican Americans may alter the genetic architecture of biomedical traits" for further consideration by *eLife*. Your revised article has been evaluated by Patricia Wittkopp (Senior Editor) and a Reviewing Editor.

The manuscript has been improved but there are some remaining issues that need to be addressed before acceptance, as outlined below:

To be able to appreciate the effects of different factors affecting complex trait variation, can the authors add the effect sizes of the important covariates to Figure 4A. Or, I would like to see these as supplemental figures. This is to put the AI ancestry effect in context, and see its magnitude relative to the effect of other non-genetic factors that have been modelled. While the authors have reported these in Supplementary file 3, figures will help more with being able to compare and parse the results. Further, I'd like to see a few sentences added to the Results and Discussion to summarize and discuss these results.

The authors have the following sentence in their Discussion "While height increases across all groups at a similar rate, illustrating the effects of non-genetic factors having an important role in the values of the trait, we do see differences based on percentage of AI ancestry." Given their new estimates of the rates of change in different groups the first part of this sentence needs to be revised.

---

## [Author Response]

Essential revisions:This paper's major finding is an observation of an increase in Amerindigenous ancestry in Mexican Americans in the 1940-1990 period. The manuscript is interesting and in theory suitable for publication in eLife, but after a number of points are addressed. First, the authors need to articulate clearly the factors that may have caused this primary observation, and what may be the most likely explanations outlined below.

We developed a simulation framework to investigate the evolutionary forces that can/cannot contribute to changes in ancestry proportions over such a short period of time. More details are included below.

Second, they need to address that the primary explanation for the ROH increase is likely the amerindigenous ancestry increase, and in that sense, determine and clearly articulate the place of the assortative mating observation in the manuscript.

We have redone the ROH analysis to control for global Amerindigenous ancestry as suggested and found that the pattern remains: ROH increases over time at a rate that exceeds the increase in Amerindigenous ancestry. In contrast, we find the opposite pattern when we perform the analogous analysis with European ancestry.

Last, they need to clearly admit in the manuscript the importance of the un-modelled socio-economic variable in the correlation between amerindigenous ancestry and complex traits, and only present this analysis after controlling for appropriate covariates.

We have removed the previous correlative analyses, and replaced them with a more thorough statistical analysis of AI ancestry with biomedical traits while controlling for several covariates. We now further discuss unmodelled socio-economic variables in the Results as well as Discussion sections.

It is a time when we are finally seeing some population genetic studies of understudied populations as they relate to complex trait variation (to which this manuscript can be an important contribution), and so the bar to be as rigorous as possible in considering alternative explanations and model all possible covariates should be set extremely high.

We completely agree that extreme care must be taken when studying marginalized populations, lest more harm may result.

1) The main finding is the increase in amerindigenous ancestry in the 1940 – 1990 period in Mexican-Americans in the United States.a) The authors state: "In our non-US born individuals (N=2987), we evaluated differences in ancestry estimates over time while accounting for years in the US and sampling weight and identified a significant effect of years in the US (𝛽=-0.0009; P=0.0006; SE=0.0003)." If I understand correctly, this regression is amerindigenous ancestry against time and other covariates. It would be helpful if the authors add to this sentence something along the lines of, "implying that individuals who arrived earlier to the US from Mexico had more European ancestry."

The reviewers understand correctly, and we have changed the sentence to reflect this: “In our non-US born individuals (N=2987), we evaluated differences in ancestry estimates over time while accounting for years in the US and sampling weight and identified a significant effect of years in the US (𝛽=-0.0009; P=0.0006; SE=0.0003) suggesting that individuals who arrived earlier to the US had less AI ancestry.”

b) Given the above analysis, and independent migration analyses (see, for example, https://www.migrationpolicy.org/article/mexican-immigrants-united-states-2) , it seems that migration from Mexico to the US shifted over the years from states with less amerindigenous ancestry to states with higher amerindigeous ancestry (Chiapas, Oaxaca, Veracruz) in the South and South-east of the country. This seems a highly likely explanation for the pattern of increasing amerindigenous ancestry that they see, and should be stated as such in the manuscript. This seems especially likely given that the signal comes primarily from non-US born individuals, or US-born individuals with parents born outside the US.

We have added to the conclusion “Independent analyses have shown that migration from Mexico to the US has shifted over the years from states with less Amerindigenous ancestry to states with higher Amerindigenous ancestry” and have cited the suggested reference.

c) The authors state that "It is possible that the increase in global Amerindigenous ancestry over time could be biased by changes in the specific subcontinental Amerindigenous ancestries over time (though such an effect is not visible in our UMAP analysis, Figure 1B)." – It is not clear what is meant by this sentence – please re-phrase and articulate more clearly. If it alludes to the difference in migration sources over time I mention above, I don't think their analyses of Fst and genetic diversity rule that explanation out.

This sentence has been simplified to “We next explored whether the increase in global Amerindigenous ancestry over time could be biased by local changes in the specific subcontinental Amerindigenous ancestries over time.”

d) Assortative mating (Figure 2D and Supplementary figure 9). This argument is puzzling because if there is assortative mating along indigenous ancestry as they suggest, then would this not mean that there is also assortative mating along the collinear European ancestry? If this is the case, why would amerindigenous ancestry be increasing in particular? The authors do state that assortative mating would not cause an increase in one ancestry. In that case, the paper overall does not provide an explanation for why the amerindigenous ancestry is actually increasing – is the migration sources explanation the most likely explanation? Along with that individuals with higher amerindigenous ancestry must be reproducing more? The likely explanations of the primary result should be made very clear for the reader.

Within Appendix 1, we have added new simulations that demonstrate how different factors such as population growth, migration, fecundity and assortative mating can shape differences in global ancestry patterns. According to the simulations, migration can have a large effect on shaping patterns of Amerindigenous ancestry, but other factors can shape these patterns as well. In particular, while assortative mating does not lead to differences in Amerindigenous ancestry over time on its own, assortative mating can amplify other factors (such as ancestry-related differences in fecundity). Given these simulations and the data we have analyzed, we argue that there is no single cause for the increased Amerindigenous ancestry over time. Rather, this increase is a result of all factors: migration, ancestry-related fecundity differences, ancestry-biased assortative mating, and population growth.

e) The authors state: "and this increase cannot be entirely explained by very recent migration." What is the evidence backing this claim?

This is based on our analyses discussed in the “Dynamic Global Ancestry Proportions in Mexican Americans” section. Specifically starting with, “In our non-US born individuals (N=2987), we evaluated differences in ancestry estimates over time while accounting for years in the US and sampling weight and identified a significant effect of years in the US (𝛽=-0.0009; P=0.0006; SE=0.0003) suggesting that individuals who arrived earlier to the US had less Amerindigenous ancestry. However, this did not change the effect of birth year on the proportion of global Amerindigenous ancestry (𝛽 = 0.0028; P<2e-16, SE=0.0003).” To clarify the conclusion, we have rephrased the sentence: “and this increase cannot be entirely explained by very recent migration based on our analyses non-US born individuals "

f) "Sampling weight" is not actually defined in the Materials and methods. Can the authors clarify how this is defined and used to weight the major analysis?

We have added the definition of the sampling weight to the “Study dataset and initial quality control” section within the Materials and methods and have cited the paper. Specifically, “The sample survey for design for HCHS/SOL has been described previously. Briefly, census block groups were selected in defined communities near each of the four recruitment centers, and households were sampled within census block groups. Households with Hispanic/Latino surnames and individuals as well as residents over 45 years old were oversampled in order to increase representation of the Hispanic/Latino target population and achieve a uniform age distribution. Sampling weights were calculated for each individual to reflect the probability of sampling.”

g) Please describe the bootstrap resampling performed – is the bootstrapping performed over individuals or segments of the genome? Please justify the strategy picked. This should be described in the manuscript.

We performed bootstrap resampling over individuals and this has been further described in within the manuscript. Specifically, we now say “Bootstrap analyses (Figure 2B and Figure 2—figure supplement 3) were performed by calculating relevant statistics based on repeated resampling of individuals with replacement. Bootstrap resampling results in an estimate of the variance of the statistics that we are calculating in our data, and allows us to assess the impact of outliers (who are only resampled in a subset of iterations).”

2) The pattern of ROH change over time observed.a) What are the number of individuals in each decade? Please show this in the manuscript. If there are more individuals in later birth decades (as may be expected), you would see an increase in the ROH summed over each genome with time, simply because you are summing over more individuals at later time periods. It is not clear if the analyses for Figure 2C are normalized by the number of individuals in each decade – if not, this would be important to do, and only the normalized results should be reported.

Supplementary file 2. We have clarified within the figure captions that the ROH sums are ROH sums per person, but to address the below comment as well, we show the normalized data in Figure 3C.

b) The ROHsum increasing with time could simply be due to the amerindigenous ancestry increasing with time, as amerindigenous ancestry carries more short ROH segments than European ancestry (see for example Ceballos et al. Nature Review Genetics 2018). The authors should explicitly describe this, as this simplest explanation would not require assortative mating to be invoked either.

We thank the reviewers for drawing our attention to this point that we missed in our original analysis. We have redone this analysis by normalizing the ROH sums per person by their global Amerindigenous ancestry. We redid the analyses with the normalized data and this is reflected now within the “Increased runs of homozygosity over time” Results section including Figure 3C and Figure 3—figure supplement 3. Our prior conclusion remains. ROH increases at a rate faster than the increase in Amerindigenous ancestry.

3) Correlation of amerindigenous ancestry and complex traitsMany of the traits studied would also be affected by socioeconomic status (for example, height, cholesterol). Do the authors have this variable available? If yes, it should be included in the multiple regression. If not, it should be clearly mentioned that they are not able to account for this likely important effect, leaving their estimates confounded by socio-economic differences that likely correlate with amerindigenous ancestry. For Figure 3, we don't think it is fair to show tau between only amerindigenous ancestry and traits as this analysis does not account for important covariates, and would like to see supplementary file 3 instead to replace Figure 3 (in a figure form as the authors prefer) such that only the multiple regression effect sizes are reported in the manuscript that account for covariates.

We agree with the reviewers, our correlation analysis was too simplistic. As suggested, we have replaced this analysis with our multiple regression model that accounts for birth year, center, sex, and sampling weight (and included all regression statistics in Supplementary file 3). While some of the particular traits that are significant after Bonferroni correction changed slightly (we are now controlling for 5*69=345 tests instead of just 69), the overall conclusion remains: nearly 1/3 of the traits are correlated with genomic Amerindigenous ancestry. While we agree that socio-economic factors can have a direct impact on biomedical traits (and can also be correlated with Amerindigenous ancestry), HCHS/SOL did not collect this variable so we cannot include it in our analysis.

Why do a first pass of this analysis without covariates included, and then re-run with covariates in the Bonferroni significant subset of traits only? (Given that there will be confounding between Amerindigenous ancestry and socioeconomic, environmental and other non-genetic factors, and especially age). Furthermore, looking in the Materials and methods section, we cannot seem to find the full description of how this analysis was performed i.e. what models were run, and how/if phenotypic measures used were cleaned and normalized etc. Please provide this.

The reviewers are entirely correct, the first-pass correlative analysis was unwarranted. As discussed above, we replaced this analysis with the multiple regression model that accounts for birth year (e.g. age), center, sex, and sampling weight. To compare the effect of AI ancestry across traits, we quantile normalized all traits, and include a justification for our use of quantile normalization in the main text. We also compared the p-values for Amerindigenous ancestry effects across traits when the data were untransformed vs quantile normalized, and found a strong correlation (rho=0.944; p<2.2e16) with no statistical evidence for a difference in the distributions of p-values (MannWhitney U test p-value=0.857).

[Editors' note: further revisions were suggested prior to acceptance, as described below.]

Summary:The manuscript is significantly improved, and the addition of the new simulation analyses greatly help in the interpretation of the trends that they see. The authors have sufficiently addressed concerns about two of their main results: (1) Interpretation of ancestry change over time and (2) Interpretation of ROH change over time. I still have the following points I would ask them to consider regarding their third main result (3) the potential effects of ancestry and ancestry change over time on the genetic architecture of complex traits, and to make appropriate additions/revisions to their analyses and text before submitting a revised manuscript. Beyond that, I see that the new simulation results don't appear in the manuscript until the Discussion. I would consider them a result and would like to see the authors try to integrate the main simulation results in the Results section, when they report their empirical observed patterns.

We appreciate the overall positive sentiment of our revision, and the focus on what additional steps would further improve our manuscript. We have now moved the simulations from Appendix 1 to the main Results section. They are now discussed after the “Strong ancestry related assortative mating in HCHS/SOL Mexicans” section and before the “Genetic association of global AI ancestry with biomedical traits” section. We hope our revisions on the potential effects of ancestry and ancestry change over time on the genetic architecture of complex traits section sufficiently address the requested revisions, as we describe further below.

Revisions for this paper:The authors state in the manuscript that, "As illustrated in Figure 4A, 20 of these traits (29%) are significantly correlated after Bonferroni correction (P<0.000145), highlighting the need for increased investigation into the role of AI genetic ancestry and other unmodelled socio-economic variables in admixed populations such as Mexican Americans." As written, it understates the correlation between AI ancestry and unmodelled socio-economic variables in the United States which we know exists on the basis of historical and social science research. Given this, I would like to see the text revised to say that it is not clear what the correlation with AI ancestry implies, and while it could be reflecting genetic effects, it can also be reflecting socio-economic variables that are correlated with AI ancestry and that AI ancestry could be serving as a proxy for. The authors themselves show that AI ancestry is correlated with educational attainment levels which they state is a proxy for socio-economic status. I would like to see their model for testing the effect of ancestry on traits include as covariates: (1) educational attainment as a proxy for socioeconomic status, (2) whether they are US born or not, and whether their parents are US born or not, to help capture the effects of different environments they would have been born in, and that their parents would have created for them on various levels. I would also encourage the authors to make their model as rich as possible by adding other environmental variables they could obtain on their recruitment sites (altitude, latitude, longitude, population density, average obesity rates to name some that are likely relevant). Ideally, this kind of analysis would be done in a mixed model framework as well, correcting for the full genomic relationship matrix, and adding a random effect to account for unaccounted for environmental factors, but they should at least add covariates to their multiple regression framework that they have access to, or could access. They should also consider issues of collinearity as they may affect their estimates and study and report the Variance Inflation Factors (VIF) of the different variables. They should report in the manuscript, the results for the full model, giving coefficients and p-values for not only AI ancestry but also the other test variables, and should describe these in the results as well, and report and discuss the contribution of other test variables relative to AI ancestry as estimated from their model.

We appreciate the reviewers’ focus on improving our statistical model. As suggested, we added covariates for educational attainment, US born status, and number of US born parents in our multiple regression as these were the variables that we had access to. These results have been updated in Figure 4A, Supplementary file 3 and Figure 4—figure supplement 3. Supplementary file 3 specifically includes the coefficients and p-values for all test variables for each trait. Here we have included a figure (Author response image 1) to illustrate the differences in the effect size of AI ancestry before and after the additional adjustments accounting for educational attainment, US born status and number of US born parents. Notably, there was very little change. This was in additional to the original adjustments of birthyear, gender, center, and sampling weight. However, even after accounting for these additional variables, the effect sizes of AI ancestry were largely unchanged (Pearson correlation coefficient *=* 0.984, P<2.2E-16). There was one trait (% Immature granulocytes) that changed from a negative association with AI ancestry to a positive association with AI ancestry, but the AI effect on this trait was not statistically significant before or after the addition of additional covariates.

We modified the above reviewer-quoted statement to “As illustrated in Figure 4A, 18 of these traits (26%) are significantly associated with AI ancestry (Bonferroni correction P<9.1E-5) after adjusting for several factors including birth year, educational attainment, US-born, and number of US-born parents. While this suggests that genetic ancestry has an effect on several traits, other unmodeled socio-economic variables that are correlated with AI ancestry may also be contributing to these patterns. Regardless, these findings highlight the need for increased investigation into the role of AI genetic ancestry in admixed populations such as Mexican Americans.”

We have included Supplementary file 3, which includes results for all phenotypes (raw and quantile normalized) with the effect size, SE, and P-value for each covariate.

Further, the authors results show three observations that I would like to see described in the Results and discussed in the Discussion, as the implications are important. The authors observe an increase in height in Mexican Americans (at roughly the same rate in all amerindigenous ancestry stratifications, see note below) with birth year. First, they do not see any trend of the polygenic height score with birth year. This suggests that while the genetic predisposition of the trait remains the same, the trait has changed significantly due to non-genetic environmental factors.

We have elaborated on this further below.

Second, even though amerindigenous ancestry is negatively correlated with height, and amerindigenous ancestry is increasing over time, the trait value/height increases rather than decreases over time. This also points to the effects of non-genetic factors playing an important role in values of the trait.

This is true. Both genetic and non-genetic factors drive variation in height. Height is estimated to have a broad-sense heritability of 80% in Northern European populations, suggesting that environmental factors explain ~20% of the variation in height in these populations. It is unclear how these estimates of heritability translate to Mexican Americans. Within the Discussion, we have elaborated further on this. We have added, “While height increases across all groups at a similar rate, illustrating the effects of nongenetic factors having an important role in the values of the trait, we do see differences based on percentage of AI ancestry. Individuals with lower percentages of AI ancestry were taller on average than individuals with higher AI ancestry pointing the role of AI ancestry on the trait.”

Last, if amerindigenous ancestry is negatively correlated with height due to genetic reasons, then shouldn't we expect to see the polygenic height score decrease with birth year, as amerindigenous ancestry increases with birth year? How do the authors interpret this meta pattern across their analyses, and what implications does it have for how temporal change in ancestry can alter the genetic architecture of traits? Overall, this could mean that (1) the correlation of amerindigenous ancestry with height is at least partly due to genetic reasons. Given that, while ancestry trends (AI ancestry increasing), combined with correlations of ancestry with traits, may make you predict one thing with respect to the genetic architecture of traits (height will decrease with birth year), the way heredity interacts with the environment through the randomness of development makes the trait move in the opposite direction than the model would predict. Or (2) the correlation of amerindigenous ancestry with height is fully picking a signal of height being lower in individuals with higher amerindigenous ancestry due to primarily environmental reasons, and therefore, as environment changes, the correlation is not meaningful for prediction. I would like to see the authors consider the above, and state these patterns as they stand across analyses, and discuss their implications for their overall thesis (as it relates to height and traits in general).

This is a very complex issue, and we have attempted to be conservative in the way we describe these patterns. For example, Figure 4—figure supplement 2 shows that there is indeed a slight negative overall trend for polygenic height score and birth year when we accounted for additional environmental variables including educational attainment, US born status and number of US born parents. The slope is not significant (P=0.14), so the approach we take is to not draw conclusions upon it. As a further point of clarification, PHS is only correlated with observed height in the bottom two quartiles of AI ancestry (i.e., only the Mexican Americans with highest European ancestry). As AI ancestry increases over time, we expect the performance of PHS to decrease. Such a decrease in accuracy could also manifest as an elimination of signal with birth-year.

It is possible we did not see a significant trend because the of the way the polygenic height score is calculated as a metric. As we know, the majority of GWASs have been performed in populations with primarily European ancestry thus providing insight into our understanding of the genetic architecture of height. However, due to the exclusion of diverse populations, we are still limited in our full understanding of the genetics of height.

A recent study of Peruvians (1) demonstrated the role of population specific variants and their contributions to differences in height. As we do not fully understand the genetics of height in Amerindigenous or admixed populations, it is possible there may be ancestral specific variants within the AI component in Mexicans. These variants may have a significant effect on the trait but these variants would not have been captured in the PHS as these GWAS results were derived from an analysis on European individuals.

Individuals with higher AI ancestry may harbor variants that may contribute more to differences in height that would not have been detected in a European GWAS. Even for Mexican individuals with higher European ancestry, they still may have variants within the AI component that could have a significant impact on the trait. However, these analyses exceed the limitations of the data available and are therefore outside the scope of this manuscript.

Note: They say in the study, "We find a similar trend in the HCHS/SOL Mexican Americans (Figure 4B). Indeed, when we stratified individuals by quartiles of global AI ancestry, we see that all quartiles have increased in height by a similar amount over the period investigated." Can the authors make this statement more specific in the manuscript – what are the rates of change in the different quartiles? Are they higher in quartiles with higher indigenous ancestry? Please integrate this into the Discussion above as well.

We have elaborated this further in the manuscript by specifically adding to the results, “The rates of change in height between AI quartiles were all positive and significant (P<5E-6). The largest was for the quartile with the highest AI ancestry, but the rates did not change monotonically with respect to AI ancestry across quartiles. The estimates for the quartiles with their 95% CIs are: 𝛽=0.135 (CI:0.097-0.173) for AI>0.58; 𝛽=0.124 (CI:0.089-0.160) for 0.46<=AI<=0.58; 𝛽=0.083 (CI:0.047-0.119) for 0.37<=AI<=0.46; and 𝛽=0.113 (CI:0.074-0.151) for AI<0.37.”

[Editors' note: further revisions were suggested prior to acceptance, as described below.]

The manuscript has been improved but there are some remaining issues that need to be addressed before acceptance, as outlined below:To be able to appreciate the effects of different factors affecting complex trait variation, can the authors add the effect sizes of the important covariates to Figure 4A. Or, I would like to see these as supplemental figures. This is to put the AI ancestry effect in context, and see its magnitude relative to the effect of other non-genetic factors that have been modelled. While the authors have reported these in Supplementary file 3, figures will help more with being able to compare and parse the results. Further, I'd like to see a few sentences added to the Results and Discussion to summarize and discuss these results.

We agree with the suggestion of a new figure and how the effects of all of the different factors can be better appreciated in addition to Supplementary file 3. Instead of adding to the main Figure 4A, we added a supplementary figure (now Figure 4—figure supplement 1) due to the high number of total points (traits x covariates). Within the Results section we have included, “While this suggests that genetic ancestry has an effect on several traits, other unmodeled socio-economic variables that are correlated with AI ancestry may also be contributing to these patterns (though AI ancestry has among the strongest effects on a range of biomedical traits, comparable to the effects of gender; Figure 4—figure supplement 1).”

For context the paragraph now reads as, “As illustrated in Figure 4A, 18 of these traits (26%) are significantly associated with AI ancestry (Bonferroni correction P<6.6E-5) after adjusting for several factors including birth year, center, gender, sampling weight, educational attainment, US-born status, and number of US-born parents. While this suggests that genetic ancestry has an effect on several traits, other unmodeled socio-economic variables that are correlated with AI ancestry may also be contributing to these patterns (though AI ancestry has among the strongest effects on a range of biomedical traits, comparable to the effects of gender; Figure 4—figure supplement 1). Regardless, these findings highlight the need for increased investigation into the role of AI genetic ancestry in admixed populations such as Mexican Americans.”

Within the Discussion, we have rephrased one of the sentences, “We identify several biomedical traits that are associated with Amerindigenous ancestry, and show that in the case of height, there are both ancestry and temporal effects” to “We identify several biomedical traits that are associated with Amerindigenous ancestry, with effects comparable to the high effects of gender, and show that in the case of height, there are both ancestry and temporal effects.”

We kept the new minor additions simple as we believe the sections in the Discussion that we had previously written about the importance of studying diverse populations are solid.

The authors have the following sentence in their Discussion "While height increases across all groups at a similar rate, illustrating the effects of non-genetic factors having an important role in the values of the trait, we do see differences based on percentage of AI ancestry." Given their new estimates of the rates of change in different groups the first part of this sentence needs to be revised.

We have reworded the sentence to “While we do see differences in mean height based on percentage of AI ancestry, height increases over time in all groups at similar rates.” We hope this clarifies that the “differences based on percentage of AI ancestry” were referring to the mean heights for each group rather than the rates.

References

1) Asgari S, Luo Y, Akbari A, Belbin GM, Li X, Harris DN, et al. A positively selected FBN1 missense variant reduces height in Peruvian individuals. Nature. 2020;582(7811):234-9.